# Neural Theorem Proving for Verification Conditions: A Real-World Benchmark

**Qiyuan Xu[1,3], Xiaokun Luan[2], Renxi Wang[3], Joshua Ong Jun Leang[4,6], Peixin Wang[5], Haonan Li[3],\*Wenda Li[6], Conrad Watt[1]**

[1] Nanyang Technological University      [2] Peking University      [3] MBZUAI
[4] Imperial College London    [5] East China Normal University    [6] University of Edinburgh

## Abstract

Theorem proving is fundamental to program verification, where the automated proof of Verification Conditions (VCs) remains a primary bottleneck. Real-world program verification frequently encounters hard VCs that existing Automated Theorem Provers (ATPs) cannot prove, leading to a critical need for extensive manual proofs that burden practical application. While Neural Theorem Proving (NTP) has achieved significant success in mathematical competitions, demonstrating the potential of machine learning approaches to formal reasoning, its application to program verification—particularly VC proving—remains largely unexplored. Despite existing work on annotation synthesis and verification-related theorem proving, no benchmark has specifically targeted this fundamental bottleneck: automated VC proving. This work introduces **Neural Theorem Proving for Verification Conditions (NTP4VC)**, presenting the first real-world multi-language benchmark for this task. From real-world projects such as Linux and Contiki-OS kernel, our benchmark leverages industrial pipelines (Why3 and Frama-C) to generate semantically equivalent test cases across formal languages of Isabelle, Lean, and Rocq. We evaluate large language models (LLMs), both general-purpose and those fine-tuned for theorem proving, on NTP4VC. Results indicate that although LLMs show promise in VC proving, significant challenges remain for program verification, highlighting a large gap and opportunity for future research.

## 1 Introduction

Program verification has been fundamental to software reliability for over half a century (Hoare, 1969). While numerous industrial program verifiers have been developed and deployed in history (Cousot et al., 2005), the adoption of program verification remains limited to safety-critical domains (Rushby, 2009; Woodcock et al., 2009). A primary reason is the heavy manual effort required in the theorem proving of *Verification Conditions (VCs)* (Barnett et al., 2006): the logical propositions that encode program correctness.

VC plays a central role in the conventional workflow of program verification (Cohen et al., 2009; Leino, 2010) as shown in Fig. 1: the Verification Condition Generator (VCG) component aims to generate VCs and the prover aims to prove them. Conventionally, VC proving is carried out by Automated Theorem Provers (ATPs). However, ATPs excel only at specific domains of problems, and require human intervention (e.g., manual proofs and annotations) when automatic proof attempts fail or time out. Taking the widely-used industry tool Frama-C (Baudin et al., 2021) as an example, existing ATPs' insufficient capability necessitates ∼600 lines of annotations for a linked list library, nearly matching the original C code length. Consequently, due to the central role of VC proving and the inadequacy of current automated approaches, VC proving has become *a key bottleneck* in automated program verification.

Large language models (LLMs) have opened the door to Neural Theorem Proving (NTP) (Minervini et al., 2018), where models generate formal proofs to conduct theorem proving. While existing NTP research has focused primarily on mathematical domains, proving competition problems (Zheng

---
\*Corresponding Author

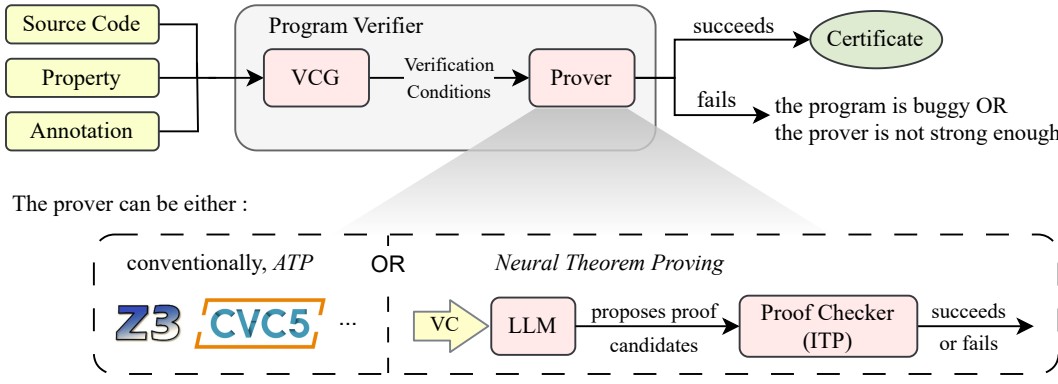

Figure 1: The conventional and NTP-based workflow of program verification.

et al., 2022; Tsoukalas et al., 2024) and formalizing mathematics (Xin et al., 2025), theorem proving extends naturally to VC proving (Harrison et al., 2014).

This motivates our central question: *can NTP automate VC proving?* To answer this, we introduce **Neural Theorem Proving for Verification Conditions (NTP4VC) — a task that applies machine-learning-based proof generation to conduct the theorem proving of VCs.**

To evaluate this task, we construct the first benchmark for NTP4VC, whose major features are compared with prior works in Tab. 1. A challenge of this construction is that Lean (de Moura & Ullrich, 2021), a mainstream language in the NTP community, has relatively fewer mature program verification frameworks built on top of it and large-scale industrial verification projects using it. Despite our best efforts, we find no sufficient native VCs available in Lean for a NTP4VC benchmark.

We overcome this issue by translating the VCs generated from other industrial verification pipelines (Why3 (Filliâtre & Paskevich, 2013) and Frama-C (Baudin et al., 2021)) into Lean. This approach also allows us to translate VCs to Isabelle (Paulson, 1990), Rocq (Coquand & Huet, 1988) (which are already implemented), and potentially other target languages, forming the first multi-language benchmark in NTP-based program verification. More crucially, this approach further allows extracting VCs from existing verification projects for industrial software, such as the Linux kernel's scheduler (Lawall et al., 2025), library functions Efremov et al. (2018), and Contiki OS's memory allocator (Mangano et al., 2017) and linked-list library (Blanchard et al., 2018).

Unlike LLM-based translation approaches that suffer from LLMs' unreliability, our translation pipeline is based on ~800 expert-written translation rules for each of the three target languages (so ~3 × 800 in total). These rules are explicitly chosen to ensure semantic preservation from the origins to the translations, thereby better ensuring the quality of the benchmark cases compared to LLM-based translation approaches.

We further evaluate several existing provers and LLMs on NTP4VC. For language-specific fine-tuned provers, the best model achieves only 2.08% pass@1, while general-purpose LLMs achieve lower performance, with GPT-o4-mini-high achieving 1.19% pass@1. These results highlight the substantial difficulty of VC proving and the need for progress in NTP and LLM reasoning.

To summarize, our contribution includes:

1. We define the task of NTP4VC (§ 1), which aims to attack the automated proving of VC, a key bottleneck in program verification.

2. We propose a *reliably automatic* method for extracting corpora from real-world verification projects (§ 3). The implementation is open-sourced.

3. We present the first real-world, multi-language benchmark for NTP4VC, with open-sourced implementation and extensive evaluation of existing provers and LLMs (§ 5).

Table 1: Comparison between our benchmark and previous ITP-based benchmarks for program verification. **VC**: the proportion of VC test cases. **Industrial pipeline**: whether the work uses industrial program verification pipelines. **Language**: the proof language supported by the benchmark.

| Benchmarks | Focus | VC | Indstrial Pipeline | Language | | |
|---|---|---|---|---|---|---|
| | | | | Lean | Isabelle | Rocq |
| Lin et al. (2024) | verificatoin-related lemmas | < 17% | ✓ | ✗ | ✓ | ✗ |
| Thompson et al. (2025) | | < 20% | ✓ | ✗ | ✗ | ✓ |
| Thakur et al. (2025) | programming puzzles in Lean | 0% | ✗ | ✓ | ✗ | ✗ |
| Dougherty & Mehta (2025) | | 0% | ✗ | ✓ | ✗ | ✗ |
| Lohn & Welleck (2024) | | 0% | ✗ | ✓ | ✗ | ✗ |
| Ours | VCs from puzzles & industrial projects | 100% | ✓ | ✓ | ✓ | ✓ |

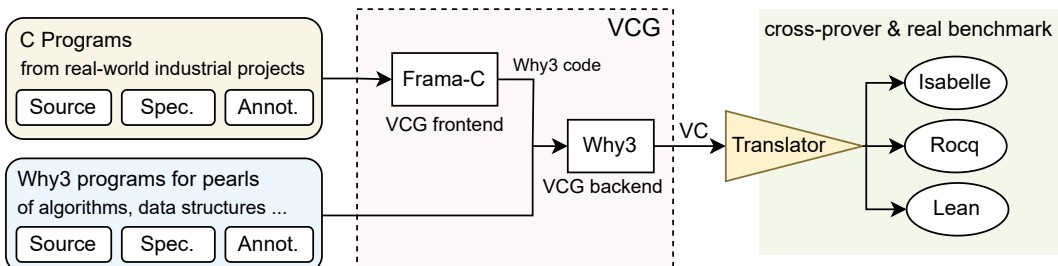

Figure 2: Our pipeline for extracting benchmark cases.

## 2 BACKGROUND

Theorem proving falls broadly into two categories: **Automated Theorem Proving (ATP)** and **Interactive Theorem Proving (ITP)**. ATP achieves full automation within specific domains of proof problems. These domains are limited, and VCs in real-world verification projects often exceed these domains, leading to proof failures and inevitable human intervention (e.g., manual proofs and annotations) in order to complete the proofs. By contrast, ITP provide highly expressive languages that enable users to construct proofs across broad domains, capable of handling almost all program verification problems. Mainstream ITP languages include Isabelle, Rocq, and Lean.

**Program verification** aims to verify that a program satisfies a given property. Ideally, a strong enough verifier should be able to complete the verification solely given the source code and the property. In practice, however, due to limitations in both VCG and VC provers, users often have to provide manual proofs and annotations to guide the verifier in completing the verification. The manual effort for these proofs and annotations constitutes a huge cost burden in program verification.

**Why3** and **Frama-C** are famous program verifiers widely used in the industry. Why3 provides 1) a language for both programming, annotation, and specifying functional correctness, 2) a VCG, and 3) powerful ATPs. A limitation is that Why3 can only verify programs written in its abstract specification language. In order to verify programs written in industrial languages, Why3 is widely used as the verification backend of well-known toolchains which translate their input language to the Why3 specification language — including Frama-C, Cameleer (Pereira & Ravara, 2021), Creusot (Denis et al., 2022), and EasyCrypt (Barthe et al., 2011). Frama-C is an industrial verifier for the C language. It provides a frontend to process C source code and then calls Why3 to complete the verification. The input of Frama-C is C source code with properties and annotations provided as comments, and the output is Why3 code that Why3 can continue to verify. Finally, Frama-C is widely used, having verified enormous industrial programs, such as air traffic management algorithm (Dutle et al., 2021), embedded operating system (Mangano et al., 2017; Blanchard et al., 2018), cryptographic modules (Peyrard et al., 2018), Linux kernel scheduler (Lawall et al., 2025), and JavaCard virtual machine (Djoudi et al., 2021).

Figure 3: The generation process of the benchmark cases and potentially training corpora.

# 3 A RELIABLE AND AUTOMATIC METHOD FOR CORPORA GENERATION

This section presents the method we use to extract real-world VCs that consitute our benchmark. The key idea is to reuse existing industrial VCGs to extract VCs from existing verification projects, and translate these VCs into the language of the target ITPs (§ 3.1). Since the projects have all passed the verifiers' checks, the VCs are guaranteed to be provable. However, this also makes them too easy to serve as valuable benchmark cases, as they may already be within reach of existing ATPs. To produce challenging benchmark cases, we introduce a novel complication process (§ 3.2) to make VCs harder while keeping them provable. The complete process is illustrated in Fig. 3.

## 3.1 VC EXTRACTION & RULE-BASED TRANSLATION

Various VC languages are used in industry, such as Why3, TPTP (Sutcliffe, 2024), and SMT Lib (Barrett et al., 2010). We adopt Why3 because its logic system is Simple Typed Theory (Church, 1940), a relatively high-level system that is close to and entailed by the logic of mainstream ITPs like Lean, Isabelle, and Rocq, ensuring the feasibility of the translation.

The translation process begins with a given Why3 source code. It first runs Why3 VCG to generate VCs and calls our customized Why3 printer to dump the VCs into an XML representation of their Abstract Syntax Trees (ASTs). These ASTs are processed by a Python translation framework also written by us and finally mapped into the target ITPs' languages. The details are provided in appendix E.

While the above process enables the basic translation from Why3 to target ITP languages, our work goes beyond this to strive for idiomatic translations that closely approximate native expressions on the target ITP platforms. For this, our translation process incorporates enhancements from two aspects: First, at the syntactic level, we use printing rules to map specific term structures to their corresponding pretty syntax defined in the ITP, including prefix, infix annotations, and ad-hoc syntax sugars like if-then-else, match-case, and list[index]. Second, we build a rewriting system to rewrite specific combinations of terms into more idiomatic expressions. Examples include rewriting integer operations into natural number operations that are more common in ITP.

The implementation of the pipeline is made of more than 2400 mapping & rewriting rules written by human experts in ITP, in total for Isabelle, Lean, and Rocq. The correctness of the rules is supported by syntax checking over the translation results on one hand, and cross-validation by other experts (our first, second, and last authors) on the other hand. These expert-written rules form the foundation of the translations' correctness and quality. Once this foundation is built, the entire translation process is automatic, constituting a *reliably automatic* method for extracting VC corpora.

## 3.2 COMPLICATION PROCESS: EXTRACTING CHALLENGING VCS

As mentioned at the beginning of this section, the VCs extracted from real-world projects are already provable by existing ATPs, thus providing insufficient challenge for benchmark evaluation. However, these VCs are provable by the ATPs as human developers have already written sufficient annotations to make them easy for ATPs to prove, rather than from inherent ATP strength. A direct idea is to erase these auxiliary annotations and restore the verification tasks to what they should ideally be in fully automated program verification.

Specifically, three sorts of annotations are dedicated to VC simplification: (1) `assert` annotation, which introduces a subgoal to ask the prover to first prove this subgoal and then use the proven subgoal as a lemma in the subsequent proofs; (2) `lemma` annotation, which explicitly introduces a global lemma so that the prover can later reference it to prove subsequent propositions; (3) annotation of lemma application, which explicitly instantiates (the free variables in) a lemma and advises

the prover to use it. All these annotations can be safely erased without affecting the VC's provability (by a strong enough prover) (Bobot et al., 2025; Correnson et al., 2025). In addition, they exhibit clear syntactic patterns enabling us to identify and erase them. Indeed, the exact job of our complication process is erasing the annotations. The results show this process effectively reduces the pass rate of Why3's strongest ATP from ~99% to ~62% on Why3's bundled examples.

## 4 NTP4VC BENCHMARK

The method discussed in § 3 enables effective extraction of real-world VCs from existing verification projects. By applying the method, we extract >7.5k VCs from various sources. From there, we carefully select 600 VCs to constitute a benchmark, with consideration for breadth, diversity, and the balance of difficulty levels as described below.

**Breadth, Diversity, and the Difficulty Level.** Real-world industrial projects certainly possess high value in a verification benchmark like ours, while at the same time, challenging algorithms and data structures are equally valuable verification targets due to their complexity. An issue is that a conflict exists between them: challenging algorithmic content is sparse in industrial project source code. If a benchmark focused solely on VCs from industrial projects, it would underrepresent algorithms and data structures. In order to balance the breadth of the verification scenarios involved, we divide our benchmark into two equal parts (50% vs 50%). (1) **Pearls of Programs** consists of minimal working programs that capture verification pain points, including algorithms, data structures, and well-known "hard nuts to crack", such as Binomial Heap, VerifyThis'24 competition, and Hillel challenge (Wayne, 2018). These programs are written in Why3's abstract specification language. (2) **Real C Verification**: VCs

Table 2: Categories of the benchmark

| Category | Number | ATP pass |
|---|---|---|
| *pearls of programs* | | |
| Algorithm | 55 | 20.00% |
| Data Structure | 73 | 19.18% |
| Calculation | 66 | 19.70% |
| Engineering | 54 | 20.37% |
| Competition | 52 | 19.23% |
| *real C verification* | | |
| Function | 81 | 23.46% |
| Loop | 81 | 24.69% |
| Memory | 74 | 24.32% |
| Invalid Arg. | 64 | 25.00% |
| Total | 600 | 22.00% |

from industrial C programs used in real-world projects, such as the memory allocator (Mangano et al., 2017) and the linked-list library (Blanchard et al., 2018) from the Contiki Operating System.

Each category is further divided into sub-categories (Tab. 2), with roughly balanced numbers of cases in each sub-category to maintain diversity. The pearl of programs consists of 1) Well-known algorithms such as sorting, string operations, searching, shortest path, and graph; 2) Data structures, including (balanced) trees, heaps, hash, and arrays; 3) Numerical and other calculations, such as arbitrary precision arithmetics, square root, exponentiation by squaring, and bitwise operations; 4) Engineering optimization tricks (e.g., in-place reversal of linked lists and N-queens by bitvector) and common engineering tasks (e.g., string padding, list element removal, space-insensitive comparison between strings, and the challenges by Wayne (2018)); 5) Cases from well-known verification competitions, e.g., VerifyThis (Ernst et al., 2019) and VSCOMP (Klebanov et al., 2011).

While the pearl of programs is organized by source programs' functionality, cases in the real C verification are categorized by the properties that VCs validate: 1) Function category verifies that programs' logical behavior meets the desired functionalities from a big-picture view, assuming the absence of runtime errors; 2) Loop category verifies loop termination, and loop invariants are established and maintained; 3) Memory category rules out the runtime error of invalid memory access; 4) Invalid Arg. checks that arguments and operands are valid. For example, the operands of multiplication do not cause arithmetic overflow, and the dividend is not zero.

Beyond breadth, we design the benchmark to balance difficulty across categories. We measure difficulty using the pass rate of Why3's strongest predefined ATP tactic, `Auto_Level_3` (AL3) (Bobot et al.). AL3 is a hybrid tactic that combines sophisticated heuristics and five industrial cutting-edge ATPs, Z3 (de Moura & Bjørner, 2008), CVC4 (Barrett et al., 2011), SPASS (Weidenbach et al., 2009), Alt-Ergo (Conchon et al., 2018), and E-prover (Schulz, 2002), such that a goal is proved once any of the ATPs proves the goal. The pass rate of AL3 then indicates the state-of-the-art of Why3 ATP over the benchmark cases, denoted as *ATP pass@n* in Tab. 2. A VC is deemed *hard* if AL3 fails to prove it, making its solution an open problem. Our design goal is to set each cat-

Table 3: Sources of cases in real C verification. LoC = Lines of C Code (comments are excluded).

| Project | # of VCs | LoC | License |
|---|---|---|---|
| Linked List Library in Contiki OS (Blanchard et al., 2018) | 78 | 833 | BSD-3-Clause |
| Memory Allocator in Contiki OS (Blanchard et al., 2018) | 21 | 145 | BSD-3-Clause |
| Selected Cases from C++ STL (Burghardt et al., 2015) | 53 | 3263 | MIT |
| X.509 Parser (Ebalard et al., 2019) | 9 | 5044 | BSD-3-Clause |
| Linux Kernel Scheduler's SWB Routine (Lawall et al., 2025) | 24 | 216 | GPLv2 |
| Linux Kernel Library Functions (Efremov et al., 2018) | 85 | 3533 | GPLv3 |
| String & Stdio Library in kLIBC (Carvalho et al., 2014) | 14 | 1220 | GPLv2 |
| Paparazzi UAV Autopilot's Math Lib (Pollien et al., 2021) | 16 | 3159 | GPLv2 |
| Total | 300 | 17413 | - |

Table 4: Statistics of involved operations. Format: $average$ $(25^{\text{th}} - 75^{\text{th}} \ percentile)$

| Operations | # of cases | # of operations | # of distinct oprs | Size | Depth | # of $\forall\exists$ |
|---|---|---|---|---|---|---|
| Integer Arith | 578 | $69.5 (10 - 73.5)$ | $5.0 (4 - 6)$ | $600.0 (133 - 643)$ | $55.8 (28 - 74)$ | $11.0 (1 - 13)$ |
| Non-Linear Arith | 118 | $10.7 (2 - 14)$ | $1.2 (1 - 1)$ | $1191 (228 - 1370)$ | $80.3 (42.5 - 115)$ | $15.2 (1 - 20)$ |
| Float Arith | 18 | $48.9 (38 - 61)$ | $7.1 (6 - 7)$ | $337 (209 - 544)$ | $73.8 (40.5 - 105.5)$ | $0.11 (0 - 0)$ |
| List, Sequence | 202 | $46.7 (8 - 62)$ | $4.1 (2 - 6)$ | $963 (198 - 1161)$ | $54.8 (24 - 69)$ | $18.6 (5 - 23)$ |
| Set, Map, Bag | 59 | $50.3 (10 - 55)$ | $3.8 (1 - 6)$ | $904 (321.5 - 1134)$ | $45.3 (28 - 55.5)$ | $26.7 (8 - 40)$ |
| Tree, String, Matrix | 26 | $64.5 (14 - 95)$ | $5.0 (4 - 6)$ | $822 (204 - 1062)$ | $45.9 (25.5 - 62)$ | $34.4 (6 - 42)$ |
| Memory | 297 | $29.5 (13 - 31)$ | $7.9 (6 - 10)$ | $357 (129 - 452)$ | $62.3 (37 - 80)$ | $5.2 (0 - 6)$ |
| Custom Datatype | 247 | $87.7 (7.5 - 94)$ | $7.0 (3 - 10)$ | $864 (193.5 - 1038.5)$ | $61.1 (25 - 103)$ | $16.3 (3 - 22)$ |
| All | 600 | $295.1 (71 - 339)$ | $24.8 (21 - 29)$ | $583 (129 - 632)$ | $54.5 (27 - 72)$ | $10.8 (1 - 12)$ |

egory's composition to target an AL3 pass rate between $20\% - 25\%$, a level that ensures sufficient open problems for advancing NTP while still allowing effective evaluation of existing approaches.

**Diversity of VC Expressions** While the previous subsection measures the diversity of the source and the purpose of the VCs, this subsection discusses the arithmetic and data structure operations involved in these VCs. We follow the taxonomic methodology conventionally used in the ATP field (Barrett et al., 2010; SMT-LIB Initiative), and categorize the operations according to the notions and the data types involved in their related reasoning. As listed in Tab. 4, the categories include integer arithmetics, non-linear arithmetic, and various common data structures. Some cases may define their custom datatypes beyond those provided in the standard libraries. This is captured by the *Custom datatype* category. Further details about this classification are given in appendix H.

For each of the categories, we count the benchmark cases that involve at least one such operation, and report the average, $25^{\text{th}}$, and $75^{\text{th}}$ percentile of: *# of operations*, the total number of occurrences of these operations; *# of unique oprs*, the number of distinct operation types in each case in each case; *size*, the number of atomic terms; *depth*, the height of the abstract syntax tree of the VCs; *# of $\forall\exists$*, the number of quantifiers occurring in each case. As presented in Tab. 4, the result shows our benchmark cases exhibit a wide distribution across different data structures and arithmetic operations, and also span VCs of varying scales within each category.

**Sources of the Benchmark Cases, and Their Licenses.** All the VCs in the benchmark are drawn from open-sourced verification projects. The pearls of programs come from the Gallery of Verified Programs Toccata Team (2025), released under the LGPL v2.1 license alongside Why3's source code. For real C verification, VCs are collected from multiple projects, as summarized in Tab. 3.

**VC Selection Process.** The 600 benchmark cases are selected from over 7.5k VCs. This subsection elaborates on the selection process. The process consists of three rounds: The first round determines the domain from which the benchmark cases will be selected; in the second round, one expert performs an initial screening to identify ~1.5k candidate cases; three experts then collaboratively evaluate each candidate in the final round to finalize the benchmark set of 600 cases.

Recall that cases in Pearls of Programs are sourced from the Toccata Team (2025)'s collection of 224 individual projects. In the first round, we select 100 projects from which all Pearl benchmark cases will be drawn, leaving the remaining 124 projects for potential use as training data. To collect

as many hard VCs as possible, we prioritize selecting projects rich in hard VCs. To measure if a VC is hard, we run Why3's AL3, and it is hard if and only if AL3 fails to solve it in 10 minutes on a 16-core workstation. For Real C Verification, we do not maintain such project-level separation, and the 8 projects are all used for benchmark cases.

In the second round, we first select all the hard VCs from the domain, totaling ~1.2k cases. Since we aim for the benchmark to have a 20–25% pass rate on the ATP baseline, we correspondingly select ~350 cases from the easy VCs to balance the candidate set at this stage. When selecting each easy VC, we check whether it is trivially provable (e.g., $\text{true} \wedge \text{true}$). To do this, we examine the VC's logical expression, the source program, related annotations, and the specifications to check that the property verified by the VC is meaningful and commonly encountered in program verification tasks.

In the final round, we apply the same evaluation method above to assess each case and refine the candidate set while additionally considering balanced coverage across the categories shown in Tab. 2. We also consider broad project coverage by selecting cases from different projects proportionally.

**Format of the Benchmark Cases.** Each benchmark case is a single VC (a single proof goal) placed individually in a theory file, and each such file contains exactly one VC. Every VC originates from a verification project and thus may contain project-specific concepts (e.g., the data type of binary tree), resulting in VCs with library dependencies. Consequently, this requires benchmark participants to be able to learn new concepts on-the-fly from the verification projects' dependency libraries.

**Dataset Contamination.** Our benchmark is generally free of data contamination concerns, despite all the source programs, properties, and annotations are public. This is because: (1) The transformation from program and property source code to VCs is complex. Even if LLMs were trained on the original source code, they cannot trivially generate VC-level concepts. In typical program verification workflows, VCs are generated only transiently and are not persistently stored or published unless done deliberately. (2) The VCs we use are derived from Why3 source code after a complication process, making most of them unprovable by existing ATPs, proofs for these VCs have never existed. (3) Even if we assume the proof details of the VCs from the original Why3 source code can leak information about the proofs of the complicated Why3 code, no leakage of the proof details is discovered despite our best efforts. This is expected, since Why3 never stores detailed proofs, but only records the ATP tools used, replaying them when proofs are needed. In fact, many ATPs do not support dumping detailed proofs at all. In summary, the risk of meaningful data contamination in our benchmark is extremely low.

**Caveat: Provability of Benchmark Cases.** All benchmark cases are drawn (after the complication process) from verification conditions in the projects that are formally verified. Methodologically, this should guarantee the provability of all benchmark cases. However, in practice, all VCs are processed through Frama-C, Why3, and our translation pipelines. Implementation bugs in any of these components may potentially render some benchmark cases unprovable. During our selection process, we checked the provability of each case to the extent possible. However, due to the inherent complexity of theorem proving, we cannot guarantee the actual provability of every case. Therefore, we must include this caveat in the paper regarding the provability of benchmark cases.

## 5    EXPERIMENTS AND EVALUATION

To evaluate the challenges posed by NTP4VC, we assess seven models, covering both general-purpose language models such as GPT-4o-mini (Achiam et al., 2024) and specialized models like DeepSeek-Prover-V2 (Ren et al., 2025). We also include ITP hammers to provide a baseline for comparison, including the hammers: Sledgehammer (Böhme & Nipkow, 2010) tool in Isabelle/HOL and CoqHammer (Czajka & Kaliszyk, 2018) in Rocq.

**Models** We evaluate both proprietary models (GPT-o4-mini (Achiam et al., 2024)) and open-source models (K2-Think (Cheng et al., 2025), DeepSeek-V3.1 (DeepSeek-AI et al., 2025), Qwen3 (Yang et al., 2025), DeepSeek-Prover-V2 (Ren et al., 2025), Goedel-Prover (Lin et al., 2025), Minilang (Xu et al., 2025)). Among them, DeepSeek-Prover-V2 and Goedel-Prover and specialized for theorem proving using Lean, while others are general-purpose reasoning models. We use 1.0 as the default temperature, and set the maximum number of tokens to $32,000$ during generation.

Table 5: Pass rates (Pass@1, Pass@4, Pass@8) of various NTP models and hammer-based automated theorem provers on the NTP4VC benchmark, evaluated across Lean, Rocq, and Isabelle.

| Model | Lean | | | Rocq | | | Isabelle | | |
|---|---|---|---|---|---|---|---|---|---|
| | P@1 | P@4 | P@8 | P@1 | P@4 | P@8 | P@1 | P@4 | P@8 |
| GPT-o4-mini-high | 0.50 | – | – | 0.00 | – | – | 1.17 | – | – |
| DeepSeek-V3.1 | 0.50 | 1.00 | 1.67 | 0.50 | 2.67 | 3.17 | 1.34 | 4.32 | 6.25 |
| Qwen3-32B | 0.33 | 0.83 | 1.17 | 0.33 | 1.17 | 1.67 | 0.74 | 2.53 | 3.42 |
| Qwen3-235B-A22B | 0.67 | 1.00 | 1.00 | 0.83 | 2.17 | 3.33 | 1.19 | 2.08 | 3.13 |
| K2-think | 0.00 | 0.00 | 0.00 | 0.67 | 0.67 | 0.67 | 0.00 | 0.00 | 0.00 |
| Goedel-Prover-V2-32B | 0.50 | 1.17 | 2.17 | – | – | – | – | – | – |
| DeepSeek-Prover-V2-671B | 1.67 | 2.17 | 3.00 | – | – | – | – | – | – |
| Minilang | – | – | – | – | – | – | 2.08 | 7.29 | 11.46 |
| CoqHammer / Sledgehammer | – | – | – | 5.67 | – | – | 18.00 | – | – |

**Metrics** Our primary evaluation metric is the *pass@n* metric. NTP models are queried multiple times for each problem, generating multiple proof attempts. A proof attempt is considered successful if it can be verified by the corresponding ITP and does not contain any fake proofs such as `admit` or `sorry`. Since hammers are mostly deterministic, we only report their pass@1 performance. GPT-o4-mini is evaluated with a single attempt per problem due to its cost, while other models are evaluated with 8 attempts per problem ($n = 8$).

**Prompts** We use zero-shot prompting for all models, providing only the problem statement and the necessary context such as definitions and previously proved lemmas. The full prompt structures are provided in Appendix F.

**Proof Verification** Our proof verification setup involves extracting the proof from the model's output and checking it within the corresponding ITP environment. We use the Lean 4.21.0, Rocq 8.20.1, and Isabelle 2024. To prevent excessively long runtimes, we set a timeout of 10 minutes for each verification attempt . Sledgehammer in Isabelle is configured to use its default ATPs and SMT solvers, including CVC4 (Barrett et al., 2011), CVC5 (Barbosa et al., 2022), Z3 (de Moura & Bjørner, 2008), E (Schulz, 2002), SPASS (Weidenbach et al., 2009), Vampire (Kovács & Voronkov, 2013), veriT (Schurr et al., 2021), and Zipperosition (Vukmirović et al., 2022). CoqHammer is configured to use all its supported ATPs, including E, Vampire, Z3, and CVC4. All proof verification is performed on a machine with an AMD Ryzen 9 7900X CPU and 64GB RAM.

## 5.1 RESULTS

The results summarized in Tab. 5 highlight the difficulty of program verification for NTP models. Across all three ITPs, our experiments demonstrate that all NTP models fail to achieve pass@8 scores above 12%. This stands in sharp contrast to their strong performance on mathematics benchmarks. For example, DeepSeek-Prover-V2 reaches 55.5% pass@1 on miniF2F, and Goedel-Prover-V2-32B reaches 88.1% pass@32. On a more challenging baseline such as Putnam-Bench, these models obtain 7.15% and 13.09% pass rates, respectively, with various attempts. This performance gap suggests that program verification requires fundamentally different reasoning capabilities than complex mathematical benchmarks.

Table 6: Number of problems solved and corresponding pass rates of NTP models and hammer-based provers on the NTP4VC benchmark, broken down by problem category.

| Category | NTP Models | | Hammers | |
|---|---|---|---|---|
| | Pass / Total | Pass Rate | Pass / Total | Pass Rate |
| Algorithm | 2 / 55 | 3.64% | 4 / 55 | 7.27% |
| Data Structure | 3 / 73 | 4.11% | 10 / 73 | 13.70% |
| Calculation | 5 / 66 | 7.58% | 8 / 66 | 12.12% |
| Engineering | 5 / 54 | 9.26% | 7 / 54 | 12.96% |
| Competition | 3 / 52 | 5.77% | 3 / 52 | 5.77% |
| *Pearls of Prog.* | 15 / 300 | 5.00% | 32 / 300 | 10.67% |
| Function | 6 / 81 | 7.41% | 25 / 81 | 30.86% |
| Loop | 3 / 81 | 3.70% | 19 / 81 | 23.46% |
| Memory | 2 / 74 | 2.70% | 18 / 74 | 24.32% |
| Invalid Arg. | 5 / 64 | 7.81% | 20 / 64 | 31.25% |
| *Real C Verif.* | 16 / 300 | 5.33% | 82 / 300 | 27.33% |
| Total | 34 / 600 | 5.67% | 114 / 600 | 19.00% |

```
lemma decompose_front_node'vc: removing the first element in an AVL tree is correctly implemented
proof
  fix d2 res
  assume pre: "case o1 of AEmpty ⇒ d2 = d ∧ res = r
    | ANode l1 d21 r2 h s ⇒ ∃res1. node_model (seq (m1 l1)) d21 (seq (m1 r2)) = Cons_d2 (seq (m1 res1)) ∧
        (0 ≤ (1 + (if hgt (m1 l1) < hgt (m1 r2) then hgt (m1 r2) else hgt (m1 l1)) ) - hgt (m1 res1) ) ∧
          (1 + (if hgt (m1 l1) < hgt (m1 r2) then hgt (m1 r2) else hgt (m1 l1))) - hgt (m1 res1) ≤ 1) ∧
                                        ...
          (-int balancing ≤ hgt (m1 res1) - hgt (m1 r) ∧ hgt (m1 res1) - hgt (m1 r) ≤ int balancing ⟶
          (1 + (if hgt (m1 res1) < hgt (m1 r) then hgt (m1 r) else hgt (m1 res1))) = hgt (m1 res)) )
```

Figure 4: An Isabelle proof generated by DeepSeek-V3.1 for a VC in the benchmark. This proof contains syntax errors, including a missing closing parenthesis and two redundant closing parentheses. The `seq` returns a tree's elements as a sequence in order; the `hgt` gives a tree's height; `balancing` is the balancing factor of AVL tree. The full example is provided in Appendix G.

By comparison, hammer-based provers show stronger results on NTP4VC: Sledgehammer reaches a pass rate of 18.00%, outperforming all the language models. By combining Sledgehammer, Minilang's language model achieves 11.46% pass rate, outperforming all other models. These results indicate that current state-of-the-art NTP models have yet to surpass the classical reasoning techniques in VC proving. Notably, although Minilang's model already incorporates Sledgehammer, it still fails to surpass Sledgehammer alone on NTP4VC. This further confirms that NTP4VC represents a novel domain for the model, distinct from the mathematical problems it excels at.

We also report the number of problems solved per category by NTP models and hammers in Table 6. The result confirms that hammers consistently outperform or match NTP models in all categories. However, the magnitude of this outperformance is non-uniform: the advantage of hammers over NTP models is substantially larger in Real C Verification tasks than in Pearls of Programming.

## 5.2 Error Analysis of NTP models

To understand the limitations of current NTPs on verification tasks, our qualitative analysis of failure cases reveals three recurring themes: syntactic errors, semantic confusion, and hallucination. More details are available in Appendix G.

**Syntactic Errors** A primary hurdle for NTPs is generating syntactically correct terms. For instance, a proof for an AVL tree VC (see Fig. 4) failed to parse due to mismatched parentheses. Correcting these purely syntactic errors allowed the term to be successfully parsed. More than 24% of generated Isabelle proofs contain syntactic errors. This highlights a key challenge of VCs: unlike typical math problems that prioritize semantic insight, VCs are often long, deeply-nested, machine-generated formulas. Their structure places extreme demands on a model's ability to maintain long-range syntactic coherence.

**Semantic and Pragmatic Confusion** A more profound failure is the model's misunderstanding of the proof paradigm itself. This is common in Lean, where models produce syntactically plausible but pragmatically incorrect code, leading to type errors. For example, they often use imperative-style assignments (e.g., $:= i_1 + i_2$) instead of declarative, tactic-based reasoning. This confusion is further evidenced by proof scripts degenerating into repetitive and meaningless tactic applications (e.g., "have $h_{16} := h_0$; have $h_{17} := h_1$ ..."), which occurs in more than 64% of Lean proofs generated by Goedel-Prover-V2-32B, one of the state-of-the-art NTP models. Even powerful models like DeepSeek-Prover-V2 exhibit this behavior, suggesting they become overwhelmed by VC complexity and resort to semantically inappropriate code, fundamentally misinterpreting the task.

**Hallucination of Non-Existent Entities** Finally, models frequently hallucinate non-existent constants, lemmas, or tactics. For instance, GPT-o4-mini often invokes a tactic called `why3`, which does not exist in Rocq, as a standalone proof for an entire VC. Similarly, many models introduce undefined constants or lemmas not found in the context or standard libraries. At least 9% of proof attempts in Isabelle failed due to these undefined entities. This demonstrates a failure to ground the generation process within the strict formal context provided by the prover.

## 6 RELATED WORKS

Prior benchmarks by Mugnier et al. (2025); Loughridge et al. (2025); Sun et al. (2024); Yang et al. (2024); Zhong et al. (2025) consider the **synthesis of annotations**: given source programs and properties, the task is to generate annotations that enable program verifiers to succeed. Like our work, they operate directly with industrial verifiers (e.g., Dafny (Leino, 2010), Verus (Lattuada et al., 2023)). Besides, they tackle the end-to-end automation problem, which offers direct practical value by reducing the manual annotation burden. However, as mentioned in § 2, an ideal verifier should not require annotations in the first place, and a stronger VC prover brings us closer to this ideal verifier. In terms of automated program verification, our NTP4VC task is complementary to annotation synthesis approaches — we propose to tackle the VC proving bottleneck directly, while they approach the problem indirectly through annotation generation (e.g., generating `assert` annotations that decompose hard VCs into simpler subgoals such that the provers can handle). Both of them are effective ways to improve automation in program verification and can be applied orthogonally.

There are also NTP benchmarks (Lin et al., 2024; Thompson et al., 2025) discussing **verification-related theorem proving**, typically consisting of proof goals collected from ITP projects about program verification engines and their applications. However, much of their work focuses on auxiliary lemmas used by program verifiers and specifications — such as those for preliminaries (e.g., arithmetic of bounded integers), programming language models (e.g., memory models), and abstract program models (e.g., binary tree algebra) — rather than VCs. In detail, no more than 17% of the test cases by Lin et al. (2024) might be VCs, and no more than 20% for Thompson et al. (2025)'s work (see appendix J for details). The gap between auxiliary lemmas and VCs is crucial because VCs are the direct theorem-proving targets that arise from program verification workflow (§ 2), while auxiliary lemmas cannot (completely) represent the theorem-proving tasks in program verification.

Besides, the Lean benchmarks by Thakur et al. (2025); Dougherty & Mehta (2025); Lohn & Welleck (2024) are also designed for program verification. These works suffer from a limitation — they do not follow the mainstream program verification methodologies adopted in the real-world industry. Lean is a specialized language with integrated verification capabilities, where the programming language itself serves as a logical reasoning language. This enables users to write Lean programs and directly verify them using the Lean system, without requiring a separate VCG for program analysis. However, program verification tasks in the real-world industry typically have to face industrial programming languages that differ substantially from logical reasoning languages. Typical industrial programming languages feature complex constructs such as mutable references, memory models, functions with side effects, and pointer arithmetics — none of which are involved in the program verification tasks examined by these benchmarks This contrast further underscores the necessity of employing industrial verification pipelines to extract VCs from real-world industrial projects for benchmark construction.

Finally, we also want to mention other NTP benchmarks involving much wider domains in theorem proving, such as the works by Yang & Deng (2019); Li et al. (2021); Lohn & Welleck (2024); Yang et al. (2023); Gauthier et al. (2021); Kaliszyk et al. (2017); Bansal et al. (2019); Huang et al. (2019); Leang et al. (2025a;b), which are also important benchmarks in NTP.

## 7 CONCLUSION

This work introduces Neural Theorem Proving for Verification Conditions (NTP4VC), presenting the first real-world multi-language benchmark for automated VC proving — a critical bottleneck in program verification. Alongside this benchmark, this work develops a reliable extraction method using expert-written translation rules and industrial verification pipelines (Why3 and Frama-C) to extract VC corpora from real-world verification projects and generate semantically equivalent VCs across Isabelle, Lean, and Rocq. Our evaluation of 600 carefully selected VCs from industrial projects reveals the substantial difficulty of this task: the strongest neural theorem provers achieve only 2.08% pass@1. Our error analysis reveals that the lengthy, deeply nested structure of VCs presents fundamentally different challenges to NTP models compared to mathematics competition problems. The benchmark and the corpora extraction method establish a foundation for advancing neural approaches to program verification, with the potential to achieve significant breakthroughs in automated program verification.

ACKNOWLEDGEMENT

Conrad Watt and Qiyuan Xu are supported by the Ministry of Education, Singapore, under its Academic Research Fund Tier 1 (NTU Tier 1, RG18/25). Conrad Watt and Qiyuan Xu are additionally supported by an NTU Nanyang Assistant Professorship Start-Up Grant. Wenda Li is supported by the AI for Math Fund from Renaissance Philanthropy and XTX Markets. Peixin Wang is supported by the National Natural Science Foundation of China (No. 92582108) and the CCF-Huawei Populus Grove Fund.

REPRODUCIBILITY STATEMENT

We have made a comprehensive artifact to ensure the reproducibility of our results and to encourage future research. The artifact contains the complete NTP4VC benchmark, the source code for our VC extraction tool, and all scripts required to replicate our experiments. It is available at:

https://github.com/xqyww123/NTP4VC

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

## A    LICENSING AND RULES OF ENGAGEMENT

### A.1    LICENSE

As listed in Tab. 3, the source projects of our benchmark cases are released under different open-source licenses, some of which are mutually incompatible (e.g., GPLv2 vs. GPLv3). However, since we do not compile these code bases into a single target, we can still distribute the data under a hybrid license. Accordingly, the VCs generated from these projects are released under the same licenses as their original source projects. The code written by us is released under the MIT License.

### A.2    PROTECTION TO PREVENT DATA LEAKAGE

To prevent data leakage, if any VC from a given `mlw` file is selected for the NTP4VC benchmark; all VCs from that `mlw` file should be protected and excluded from training. In contrast, other `mlw` files provided in our associated artifact are not protected, as long as no VC from those files is included in the NTP4VC benchmark. Note that the `mlw` files in the Real C Verification category are generated from C projects, with each `mlw` file corresponding to a single C function. The identity of an `mlw` file should be considered bound to its original C function. This means that `mlw` files generated through alternative means (e.g., different compilation flags) that produce varied content should also be subject to the above protection.

### A.3    ACCEPTABLE IMPROVEMENTS TO THE TRANSLATION PIPELINE

The purpose of this benchmark is to advance Neural Theorem Proving capabilities on proving Verification Conditions. Note that NTP models built on top of specific ITP platforms are inevitably influenced by the language features and idioms of their respective platforms. The rule-based translation provided in this work may not fully capture all such features and idioms, resulting in some VCs being translated into representations that are unnatural or unconventional for the target ITP platform. These should be considered as deficiencies in the translation pipeline. Therefore, to allow for addressing these potential issues, improvements to the translation pipeline that address such unnatural or unconventional representations should be considered legitimate and acceptable. To be clear, regenerating the NTP4VC benchmark using an improved translation pipeline as described above and evaluating on the regenerated benchmark is likewise considered legitimate and acceptable. However, any evaluation based on a regenerated NTP4VC benchmark must explicitly report any modifications made to the translation pipeline when presenting results.

## B    ADDITIONAL BACKGROUND

Although VC has been introduced in § 1, given its significance to our work, we provide a precise definition as follows.

**Definition 1.** Given a program and a property, a *Verification Condition (VC)* is a mathematical proposition that, when proven true, guarantees the program satisfies the desired property.

Additionally, another aspect that remains unspecified in the main text is the target property of the program verification discussed in our work. Any program verification task always considers a target property. The **target property** considered in our benchmark is **Functional Correctness**, which guarantees a program correctly implements its desired function — for any allowed input, the output of the program always satisfies a separately written logical specification of the program's behaviour (see appendix C for a concrete example). Functional correctness is a verification goal widely adopted in real-world industrial practice (Garavel et al., 2020), and it is also a primary capability of our toolchain components Why3 and Frama-C.

**Define** $\mathcal{I}(l, u) \triangleq 0 \le l \wedge u < \text{length}(a) \wedge (\forall i.\ 0 \le i < \text{length}\ a \wedge a[i] = v \longrightarrow l \le i \le u)$

```
1   exception NF (* standing for not found *)
2   let binary_search (a: array int) (v: int) : int
3       requires ∀i j. 0 ≤ i ≤ j < length(a) ⟶ a[i] ≤ a[j]
4       ensures  0 ≤ result < length(a) ∧ a[result] = v
5       raises   NF ⟶ ∀i. 1 ≤ i < length(a) ⟶ a[i] ≠ v
6   = let ref l = 0 in
7       let ref u = length a - 1 in
8       while l <= u do
9           invariant I(l, u)
10          variant u − l
11          let m = l + div (u - l) 2 in
12          if a[m] < v then
13              assert ∀i. l ≤ i < m + 1 ⟶ a[i] < v
14              l := m + 1
15          else if a[m] > v then u := m - 1
16          else return m
17      done;
18      raise NF
```

$\forall u\ l.$

$\quad \mathcal{I}(l, u) \wedge \text{sorted}(a) \longrightarrow$

$\quad\quad \text{if } l \le u \text{ then let } m = l + (u - l)/2 \text{ in}$

$\quad\quad\quad 0 \le m < \text{length}\ a\ \wedge$

$\quad\quad\quad (\text{ if } a[m] < v$

$\quad\quad\quad\quad \text{then } (\forall i.\ l \le i < m + 1 \longrightarrow a[i] < v) \longrightarrow$

$\quad\quad\quad\quad\quad 0 \le u - l \wedge u - (m + 1) < u - l$

$\quad\quad\quad\quad\quad \wedge\ \mathcal{I}(m + 1, u)$

$\quad\quad\quad\quad \text{else if } v < a[m]$

$\quad\quad\quad\quad \text{then } 0 \le u - l \wedge m - 1 - l < u - l$

$\quad\quad\quad\quad\quad \wedge\ \mathcal{I}(l, m - 1)$

$\quad\quad\quad\quad \text{else } 0 \le m < \text{length}\ a \wedge a[m] = v)$

$\quad\quad \text{else } (\forall i.\ 0 \le i \wedge i < \text{length}\ a \longrightarrow a[i] \ne v)$

Figure 5: (Left) A Why3 program for binary search, with the functional correctness property in cyan and annotations in orange. (Right) One of the generated VCs for its functional correctness (simplified).

## C  AN EXAMPLE OF VC

This section presents an example Why3 program and its VC to provide readers with a concrete sense of how VCs relate to traditional mathematical theorems.

The left side of Fig. 5 presents a Why3 program for binary search. Its functional correctness property is given by the requires, ensures, and raises clauses. requires specifies the domain of valid inputs, i.e., the given array $a$ must be sorted. ensures and raises specify the expectation of the output — conditions that the $result$ has to satisfy, which are, 1) the $result$ is a valid index (i.e., between $0$ and the length) such that array $a$'s element at the index has a value of $v$, if no exception raises, or 2), if exception NF raises, no element in the array has a value of $v$.

This program involves mutable references and an effectful loop, which makes direct reasoning with ITPs extremely tedious. The mature academic and industrial solution is to apply a specialized program reasoning engine, like Why3's VCG, to first extract pure logical proof goals, so-called VCs.

The invariant and variant clauses are annotations that help the VCG to work. The invariant clause declares a loop invariant, which is a formula that remains true throughout every loop iteration, and is required by the VCG process. The variant clause declares a metric which is strictly decreasing in each loop iteration. It helps to generate the VCs for ensuring loop termination.

The assert at line 13 is an annotation to ease the burden of VC prover. It introduces a subgoal and instructs the verifier to first prove this subgoal and then use the proven subgoal as a premise (as shown in pink in Fig. 5) in the subsequent proofs. Essentially, it helps the prover to decompose VCs into simpler subgoals.

The right side of Fig. 5 is one of the generated VCs for the functional correctness, a mathematical statement that encodes the logic behind the program's behaviors. First, invariant $\mathcal{I}(l, u)$ represents that $l, u$ are valid boundaries of the indices of the elements of value $v$. Then, consider the case of $l \le u$, where the VC verifies the loop iterations: if either $a[m] < v$ or $a[m] > v$, the updated boundary $(m + 1, u)$ or $(l, m - 1)$ must preserve the invariant, and the metric $u - l$ must strictly decrease; if the program exits and returns $m$ at line 16, the VC judges whether the return value $m$ satisfies the expectation as stated in the ensures clause, by replacing the $result$ variable in the ensures clause with $m$. At last, the last line in the VC corresponds to line 18, where the VC checks value $v$ does not appear in array $a$.

Finally, we must emphasize that this VC is simplified for better readability. The original VC is much more complicated (as shown in Fig. 6), where the invariant $\mathcal{I}$ is not defined as a term, duplicated terms abound, and $\wedge$-connected terms are disordered. This binary search is also one of the simplest cases in program verification, while other VCs can be much more complicated. This represents a gap between competition-style mathematical theorems and VCs: the former are concise but re-

```
1  exception Not_found
2
3  let binary_search (a: array int) (v: int) : int
4    requires ∀i j.  0 ≤ i ≤ j < length(a) ⟶ a[i] ≤ a[j]
5    ensures   0 ≤ result < length(a) ∧ a[result] = v
6    raises    Not_found ∧ ∀i. 0 ≤ i < length(a) ⟶ a[i] ≠ v
7  = let ref l = 0 in
8    let ref u = length a - 1 in
9    while l <= u do
10     invariant 0 ≤ l ∧ u < length(a)
11     invariant ∀i. 0 ≤ i < length(a) ∧ a[i] = v ⟶ l ≤ i ≤ u
12     variant  u - l
13     let m = l + div (u - l) 2 in
14     if a[m] < v then
15       l := m + 1
16     else if a[m] > v then
17       u := m - 1
18     else
19       assert  a[m] = v
20       return m
21   done;
22   raise Not_found
```

$$(\forall i\,j.\, 0 \leq i \leq j < \mathrm{length}\,a \longrightarrow a[i] \leq a[j]) \longrightarrow$$

$\mathrm{let}\ o_1 = \mathrm{length}\,a - 1\ \mathrm{in}$

$(0 \leq 0 \wedge o_1 < \mathrm{length}\,a)$

$\wedge\ (\forall i.\ 0 \leq i < \mathrm{length}\,a \longrightarrow a[i] = v \longrightarrow 0 \leq i \leq o_1)$

$\wedge\ \big(\forall u\,l.$

$\quad (0 \leq l \wedge u < \mathrm{length}\,a)$

$\quad \wedge\ (\forall i.\ 0 \leq i < \mathrm{length}\,a \longrightarrow a[i] = v \longrightarrow l \leq i \leq u)$

$\longrightarrow \mathrm{if}\ l \leq u$

$\mathrm{then\ let}\ m = l + (u - l)/2\ \mathrm{in}$

$(0 \leq m \wedge m < \mathrm{length}\,a)\ \wedge$

$(\ \mathrm{if}\ a[m] < v$

$\quad \mathrm{then}\ (0 \leq u - l \wedge u - (m+1) < u - l)$

$\quad\quad \wedge\ (0 \leq m + 1 \wedge u < \mathrm{length}\,a)$

$\quad\quad \wedge\ (\forall i.\ 0 \leq i < \mathrm{length}\,a \wedge a[i] = v$

$\quad\quad\quad \longrightarrow m + 1 \leq i \wedge i \leq u)$

$\quad \mathrm{else}\ (0 \leq m \wedge m < \mathrm{length}\,a)$

$\quad\quad \wedge\ (\mathrm{if}\ v < a[m]$

$\quad\quad\quad \mathrm{then}\ (0 \leq u - l \wedge m - 1 - l < u - l)$

$\quad\quad\quad\quad \wedge\ (0 \leq l \wedge m - 1 < \mathrm{length}\,a)$

$\quad\quad\quad\quad \wedge\ (\forall i.\ 0 \leq i\ \mathrm{length}\,a \wedge a[i] = v$

$\quad\quad\quad\quad\quad \longrightarrow l \leq i \wedge i \leq m - 1)$

$\quad\quad\quad \mathrm{else}\ (0 \leq m < \mathrm{length}\,a) \wedge a[m] = v))$

$\mathrm{else}\ (\forall i.\ 0 \leq i < \mathrm{length}\,a \longrightarrow a[i] \neq v)\big)$

Figure 6: The original program and the original VC of Fig. 5, without simplification

quire sophisticated mathematical skills to construct paths towards proofs, whereas VCs require less intellectual creativity, but are complicated and require the prover to process enormous formulas, potentially extracting key information from noise to simplify the proof goals and ultimately complete the proofs.

## D  LIMITATION & MITIGATION

From a methodological perspective, our VC extraction method ensures all obtained VCs are provable by construction. However, implementation bugs may occur in Why3, Frama-C, or our translation pipeline, potentially rendering some VCs unprovable. To address such potential invalidation, we design the benchmark to be updatable: we will repair the VC extraction pipeline and refresh the benchmark when invalidation occurs. Since the intended semantics of VCs are grounded in the source verification projects, these updates primarily address representation issues while preserving the essential semantics of the verification problems. However, should an invalid benchmark case be irreparable in rare instances, we will eliminate it from the benchmark to guarantee all remaining cases are provable.

## E  DETAILED EXTRACTON PIPELINE

This section provides further details on our extraction pipeline from two perspectives: approach and implementation

### E.1  METHODOLOGY DETAILS

The translation process begins with a given Why3 source code. The process first runs Why3 VCG to generate VCs and calls our customized Why3 printer to dump the VCs into an XML representation of their Abstract Syntax Trees (ASTs). These ASTs are processed by a Python translation framework also written by us and finally mapped into the target ITPs' languages.

A verification project typically contains multiple VCs that depend on shared Why3 theories consisting of lemmas, axioms, functions, and datatype definitions. These theories may further depend on

others, forming a complex dependency graph across the project. To successfully translate the VCs, we must translate all dependent theories. Our translation process, therefore, recursively handles every theory in this dependency graph, mapping the entire verification project into the target ITPs.

In terms of structure, a Why3 theory is a sequence of declarative elements consisting of axioms, definitions of functions, and algebraic data types. All three sorts of declarations have similar counterparts in the target ITPs and can be mapped to them, despite two minor gaps. One is regrading the non-uniform data type (Blanchette et al., 2017), which is not natively supported by Isabelle. Therefore, we circumvent all VCs involving such data types. The other gap pertains to discharging the termination check of recursive function definitions, a conventional requirement for ITPs to ensure the soundness of their logics. Some ITPs' termination checkers (Isabelle and Rocq) are not strong enough to automatically prove the well-foundedness of certain complicated recursions, even though Why3 has checked all the termination. Since the proof obligation of the termination is irrelevant to the semantics of VCs' proof obligation, we trust Why3's termination check and axiomatize this in the ITP translation in case ITP's termination checker fails.

Having the theory dependencies and theory-level declarations translated, the last work is to translate the term language. Both Why3's and the ITPs' term languages are based on the lambda calculus, a core language involving only variables, constants, applications, and function abstractions. This similarity simplifies a lot of the translation process. Overall, the process maps Why3 constants to the target ITPs' constants, and preserves all other variables, application, and function structures. One exception unsupported by Isabelle is Why3's add-on feature, the `as`-binding used in pattern matching, which annotates a sub-pattern with a variable and binds the term captured by this sub-pattern to the variable. We convert this into semantically equivalent let-bindings.

### E.2 IMPLEMENTATION DETAILS

The implementation of the VC extraction and translation pipeline consists of six main components:

1. A Why3 patch to export Why3's internal Abstract Syntax Tree (AST) into an XML representation (in ∼200 lines of OCaml).

2. A Python parser to read the XML representation into an S-expression representation of an extended simply-typed lambda calculus (in ∼160 lines of Python).

3. Python library functions providing basic support for manipulating the lambda calculus, such as substitution, variable deconfliction, rewriting, and folding over atomic terms (in ∼800 lines of Python).

4. A Python module for managing Why3 sessions, managing translation contexts (e.g., allocated constant/variable names in the context), and chaining all the components together to run them automatically (in ∼500 lines of Python).

5. Translation rules, rewriting rules, ad-hoc term rewriting procedures, package management, and syntax check adapter, for each of the Isabelle, Lean, and Rocq (in ∼800/790/770 lines of YAML, ∼930/780/970 lines of Python, for Isabelle, Lean, Rocq, respectively).

6. ITP libraries that map Why3 notions into the ITPs' native builtins (in ∼500/160/200 lines of Isabelle/Lean/Rocq, respectively)

The subsection elaborates on some of the nontrivial components as follows.

**The Why3 patch** is modified from Why3's existing Isabelle printer, which exports Why3 AST in XML format but with Isabelle-specific adaptations. We neutralize these adaptations to make it output the raw Why3 internal AST. Specifically, we remove its mapping from Why3 terms to Isabelle terms; add Rocq and Lean keywords to the blacklist of variable names; fix its escaping of XML special characters; add support for the as-binding syntax in pattern matching; add type annotations to definition exports.

**The S-expression** used in our internal process is a simply-typed (HOL style) lambda calculus extended with native AST nodes for finite Cartesian products, pattern matching (the `case` statement), literal numbers and strings, and the `as`-bindings (which bind the sub-term that matches a sub-pattern to a variable, in a usual pattern matching). Bound variables are represented in the same way as free

variables; we do not use De Bruijn indices, but instead maintain contextual variables and decon-
flict names of bound variables explicitly (because it simplifies our parsing and printing work, while
computational efficiency can be compromised in our context).

The **substitution**, **variable deconfliction**, and **folding** are all standard. We use Python's func-
tional programming features to implement these operations. The **rewriting** system is simplified such
that 1) all reducible expression (redex) patterns have the form (`contant` $arg_1 \cdots arg_n$) where all
$\{arg_i\}_{1 \le i \le n}$ are free variables and the arity $n$ is schematic; 2) no lambda abstraction is allowed
to appear in the contractum, so the contracta can only be atoms or (nested) function applications.
This simplification allows representing a rewriting rule as merely a tuple of the redex's constant
name, the constant's arity, and a list-represented S-expression for the contractum. We use YAML's
dictionary datatype to represent a set of rewriting rules, e.g., (`Why3.length, 1, [Int.int,`
`[Isabelle.length, ` $arg_0$`]]`) rewrites (Why3.length $l$) into Int.int (Isabelle.length $l$), for
any $l$. This greatly simplifies the writing of rewriting rules. For more complex rewritings that re-
quire more complex redex patterns, we use hard-coded Python `match-case` to work over the
S-expression directly.

## F  PROMPTS

Our work employs two types of prompts: general prompts designed for broad-purpose LLMs and
specialiZed prompts tailored for particular fine-tuned models.

The templates of the general prompts are shown as follows.

---

**General Prompt for Isabelle**

Given the following Isabelle theories as context, prove the Isabelle proposition given at the
end.

File 'NTP4Verif.thy':
{content of the theory file}

*And many other libraries* $\cdots\cdots$
File 'imp_SymStateSet.thy':
{content of the theory file}

Given the context above, consider the proposition in the following Isabelle code:
{the target proof goal together with its contextual theory}

Response the Isabelle proof only. Do not repeat any context nor the statement.

---

**General Prompt for Lean**

Given the following Lean 4 theories as context, prove the Lean 4 proposition given at the end.

File 'Base.lean':
{content of the theory file}

*And many other libraries* $\cdots\cdots$
File 'SymStateSet.lean':
{content of the theory file}

Given the context above, consider the proposition in the following Lean 4 code:
{the target proof goal together with its contextual theory}

Response the Lean 4 proof only. Do not repeat any context nor the statement.

---

---

**General Prompt for Rocq**

Given the following Rocq theories as context, prove the Rocq proposition given at the end.

File 'Base.v':
{content of the theory file}

*And many other libraries* · · · · · ·

File 'SymStateSet.v':
{content of the theory file}

Given the context above, consider the proposition in the following Rocq code:
{content}

Response the Rocq proof only. Do not repeat any context nor the statement.

---

The template specifically for Goedel-Prover and DeepSeek-Prover is as follows.

---

**Prompt for SpecialiZed Models**

Complete the following Lean 4 code:
{the target proof goal together with its contextual theory}

Before producing the Lean 4 code to formally prove the given theorem, provide a detailed proof plan outlining the main proof steps and strategies.
The plan should highlight key ideas, intermediate lemmas, and proof structures that will guide the construction of the final formal proof.

---

## G    FAILURE CASES

To investigate the failure modes of NTP models on verification conditions, we analyzed the error logs and proof scripts from our evaluation. We highlight three dominant categories of errors: syntactic invalidity, semantic degeneration, and hallucination. It is important to note that the statistics presented below represent *conservative lower bounds*. For syntax and hallucination errors, proof assistants abort execution at the first error; thus, a single proof might contain multiple subsequent errors that remain uncounted. Similarly, our detection of semantic degeneration relies on rigid regular expressions for some common cases, likely missing more subtle forms of degeneration.

Generating syntactically well-formed terms remains a primary hurdle, particularly for complex nested expressions in VCs. In our analysis of Isabelle proof attempts, we found that **at least 24%** failed solely due to syntax errors. Listing 1 shows the complete erroneous proof generated by DeepSeek-V3.1 for proving the correctness of the `decompose_front_node` function on AVL trees. This function is responsible for decomposing the front node of an AVL tree, and its correctness is specified by the corresponding VC. Specifically, the term `seq (m1 l)` represents the sequence of elements in the left subtree `l`, `d` refers to the data element of the current node, and `hgt (m1 r)` denotes the height of the right subtree `r`. The generated proof attempts to first introduce the universally quantified variables `d2` and `res`, followed by a case analysis on `o1`, which represents the structure of the AVL tree. However, the term cannot be parsed due to two subtle syntax errors: (1) a missing closing parenthesis in a deeply nested arithmetic expression on line 16, and (2) two extraneous closing parentheses on lines 18 and 25, respectively. In fact, if one only removes the last extraneous closing parenthesis, the term can be parsed. However, it will result in a term in the form of "$\cdots \wedge (0 \leq (1 + expr) - \texttt{hgt}(\texttt{m1 res1}) \wedge \cdots$", which is syntactically valid but semantically incorrect (the height of `res1` is being conjoined with another inequality). What one would expect is instead "$\cdots \wedge (0 \leq (1 + expr) - \texttt{hgt}(\texttt{m1 res1})) \wedge \cdots$", which requires removing the extraneous parenthesis on line 16 and adding a closing parenthesis after "`res1`". The lengthy logical formulas with deeply nested constructs is a common pattern in VCs, which poses significant challenges for NTP models to maintain long-range syntactic coherence.

```
1  lemma decompose_front_node'vc:
2    fixes l :: "'a t2"
3    fixes r :: "'a t2"
4    fixes o1 :: "'a view"
5    fixes d :: "'a t1"
6    assumes fact0: "-int balancing ≤ hgt (m1 l) - hgt (m1 r)"
7    assumes fact1: "hgt (m1 l) - hgt (m1 r) ≤ int balancing"
8    assumes fact2: "case o1 of (AEmpty :: 'a view) ⇒ hgt (m1 l) = (0 :: int) ∧ ..."
9    shows "case o1 of (AEmpty :: 'a view) ⇒ True
10                   | ANode l1 d2 r2 _ _ ⇒(((0 :: int) ≤ hgt (m1 l) ∧ ...))"
11   and "∀(d2 :: 'a t1) (res :: 'a t2).
12   (case o1 of (AEmpty :: 'a view) ⇒ d2 = d ∧ res = r
13             | ANode l1 d21 r2 _ _ ⇒ (∃(res1 :: 'a t2). (node_model (seq (m1 l1)) ...)))"
14 proof -
15   {
16     fix d2 res
17     assume pre: "case o1 of AEmpty ⇒d2 = d ∧ res = r
18     | ANode l1 d21 r2 h s ⇒∃res1. node_model (seq (m1 l1)) d21 (seq (m1 r2)) = Cons d2 (seq (m1 res1)) ∧
19           (0 ≤ (1 + (if hgt (m1 l1) < hgt (m1 r2) then hgt (m1 r2) else hgt (m1 l1))) - hgt (m1 res1) ∧
20            (1 + (if hgt (m1 l1) < hgt (m1 r2) then hgt (m1 r2) else hgt (m1 l1))) - hgt (m1 res1) ≤ 1) ∧
21           seq (m1 res) = node_model (seq (m1 res1)) d (seq (m1 r)) ∧
22           (0 ≤ (1 + (if hgt (m1 res1) < hgt (m1 r) then hgt (m1 r) else hgt (m1 res1))) - hgt (m1 res) ∧
23            (1 + (if hgt (m1 res1) < hgt (m1 r) then hgt (m1 r) else hgt (m1 res1))) - hgt (m1 res) ≤ 1) ∧
24           (-int balancing ≤ hgt (m1 res1) - hgt (m1 r) ∧ hgt (m1 res1) - hgt (m1 r) ≤ int balancing ⟶
25            (1 + (if hgt (m1 res1) < hgt (m1 r) then hgt (m1 r) else hgt (m1 res1))) = hgt (m1 res)))"
26
27     show "node_model (seq (m1 l)) d (seq (m1 r)) = Cons d2 (seq (m1 res)) ∧
28           (0 ≤ (1 + (if hgt (m1 l) < hgt (m1 r) then hgt (m1 r) else hgt (m1 l))) - hgt (m1 res) ∧
29            (1 + (if hgt (m1 l) < hgt (m1 r) then hgt (m1 r) else hgt (m1 l))) - hgt (m1 res) ≤ 1)"
30     proof (cases o1)
31       case AEmpty
32       with pre have d2_def: "d2 = d" and res_def: "res = r" by simp_all
33       from fact2[unfolded AEmpty] have seq_l: "seq (m1 l) = []" and hgt_l: "hgt (m1 l) = 0" by simp_all
34
35       show ?thesis unfolding d2_def res_def
36         by (simp add: seq_l node_model_def, auto simp: hgt_l fact0 fact1)
37     next
38       case (ANode l1 d21 r2 h s)
39       with pre obtain res1 where
40         decomp: "node_model (seq (m1 l1)) d21 (seq (m1 r2)) = Cons d2 (seq (m1 res1))" and
41         res_seq: "seq (m1 res) = node_model (seq (m1 res1)) d (seq (m1 r))"
42         by auto
43
44       from fact2[unfolded ANode] have seq_l:
45         "seq (m1 l) = node_model (seq (m1 l1)) d21 (seq (m1 r2))"
46         by simp
47
48       show ?thesis
49         by (simp add: seq_l decomp res_seq node_model_def,
50             metis (mono_tags, lifting) fact2 ANode case_prod_conv)
51     qed
52   }
53 qed
```

Listing 1: The complete proof generated by DeepSeek-V3.1 for the correctness of the `decompose_front_node` function on AVL trees.

```lean
1  lemma goal10 (a : Memory.addr) (t_1 : Memory.addr -> Z) (t_4 : Memory.addr -> Memory.addr) (t : Z -> Z)
2  (t_3 : Memory.addr -> Z) (t_2 : Memory.addr -> Z) :
3    let a_1 : Memory.addr := Memory.shift a (1 : Z);
4    let x : Z := t_1 a_1;
5    let a_2 : Memory.addr := Memory.shift a (0 : Z);
6    let x_1 : Z := t_1 a_2;
7    let x_2 : Z := x * x_1;
8    let a_3 : Memory.addr := Memory.shift a (2 : Z);
9    let a_4 : Memory.addr := Memory.shift a (3 : Z);
10   let a_5 : Memory.addr := t_4 a_3;
11   Memory.region (Memory.addr.base a) <= (0 : Z) -> -(2147483648 : Z) <= x_2 -> x_2 <= (2147483647 : Z) ->
12   Memory.framed t_4 -> Memory.linked t -> Memory.sconst t_3 -> Cint.is_uint16 x -> Cint.is_uint16 x_1 ->
13   Axiomatic13.p_valid_memb t t_4 t_1 a -> Memory.valid_rd t a_3 (1 : Z) -> Memory.valid_rd t a_4 (1 : Z) ->
14   Memory.valid_rd t a_1 (1 : Z) -> Memory.valid_rd t a_2 (1 : Z) -> Axiomatic12.p_valid_or_empty t a_5 x ->
15   A_MemSet.l_memset (Memory.havoc t_2 t_3 (Memory.shift a_5 (0 : Z)) x) a_5 (0 : Z) x = true ->
16   Axiomatic12.p_valid_or_empty t (t_4 a_4) (Cint.to_uint32 x_2)
17   :=
18   intro a_1 x a_2 x_1 x_2 a_3 a_4 a_5 h_region h_x2_lower h_x2_upper h_framed t_4_linked t_3_sconst h_x_uint16
19   h_x1_uint16  h_p_valid_memb h_valid_rd_a3 h_valid_rd_a4 h_valid_rd_a1 h_valid_rd_a2 h_p_valid_or_empty_a5
20   h_memset
21   have h_main : Axiomatic12.p_valid_or_empty t (t_4 a_4) (Cint.to_uint32 x_2) := by
22     have h1 : Axiomatic12.p_valid_or_empty t (t_4 a_4) (Cint.to_uint32 x_2) := by
23       -- Use the assumption `h_framed` to deduce the required property
24       -- This step is hypothetical and depends on the actual definition of `Axiomatic12.p_valid_or_empty`
25       -- and how it interacts with `Memory.framed`.
26       -- In practice, we might need more specific information or lemmas to make this step rigorous.
27       -- For the sake of this example, we suppose that `h_framed` directly implies the conclusion.
28       exfalso
29       have h2 := h_framed
30       have h3 := t_4_linked
31       have h4 := t_3_sconst
32       have h5 := h_p_valid_memb
33       have h6 := h_valid_rd_a3
34       have h7 := h_valid_rd_a4
35       have h8 := h_valid_rd_a1
36       have h9 := h_valid_rd_a2
37       have h10 := h_p_valid_or_empty_a5
38       have h11 := h_memset
39       simp_all [Axiomatic12.p_valid_or_empty, Axiomatic13.p_valid_memb, Memory.framed,
40               Memory.linked, Memory.sconst]
41       <;>
42       (try contradiction) <;>
43       (try norm_num at *) <;>
44       (try aesop)
45       <;>
46       (try
47         {
48           simp_all [Cint.is_uint16]
49           <;>
50           norm_num at *
51           <;>
52           omega
53         })
54     exact h1
55   exact h_main
```

Listing 2: Example of semantic degeneration: Redundant variable renaming in a Lean proof.

NTP models frequently lose track of the proof state, resulting in repetitive, meaningless steps. We detected this behavior by matching patterns of continuous "renaming" (e.g., using `have h1 := h2` where both `h1` and `h2` are simple identifiers) repeated at least three times. In Lean, **more than 64%** of proofs generated by Goedel-Prover-V2-32B exhibited this specific degeneration pattern. Listing 2 exemplifies the generation of repetitive and meaningless tactic applications in Lean. The model (Goedel-Prover-V2-32B) engages in a redundant "renaming ritual" (`have h2 := h_framed`, etc.), erroneously assuming that automated tactics like `simp_all` require local variable aliases to access the context. This behavior likely stems from domain shift, where the proof context is more complex than the standard mathematical corpora used for training. Furthermore, the comments (e.g., "assume that `h_framed` directly implies the conclusion") explicitly admit that the logical step is hypothetical. This suggests its inability to derive the necessary lemmas to complete the proof.

Models often invoke non-existent constants, lemmas, or tactics due to hallucinations. For instance, GPT-o4-mini frequently attempts to solve Rocq VCs using a `why3` tactic, which does not exist in the language. In Isabelle, **at least 9%** of failures were triggered by references to undefined constants or lemmas that are absent from the context. We identified these cases by explicitly matching keywords such as "Undefined fact" or "Undefined constant" in the error logs. Crucially, since the proof assistant terminates the checking process at the first encountered error, hallucinations present in the latter parts of proof scripts — especially those already halted by syntax errors or earlier tactic failures — remain uncounted. Consequently, this 9% figure represents a highly conservative lower bound.

## H  CLASSIFICATION & METRIC DETAILS OF TABLE 4

The operation classification is conducted on the Isabelle version of our benchmark. We developed Isabelle extensions to analyze the expressions of the obtained proof goals. We elaborate on the constitution of each category in Tab. 4 as follows.

- *Integer Arith* consists of addition, subtraction, multiplication, division, exponentiation, comparison, square root, and factorial operations whose operands are integers, natural numbers, or bounded integers (machine integers); and also bit-width conversions and bitwise operations.

- *Non-linear Arith* consists of multiplication, division, and exponentiation between non-constant expressions, following de Moura & Bjørner (2008) and Z3 (2025). In Tab. 4, most cases contain one distinct non-linear operator — multiplication — and sometimes contain additional operators like division and exponentiation.

- *Float Arith* consists of arithmetics on floating-point numbers. All floating-point numbers in our benchmark cases are modeled as real numbers, disregarding precision errors.

- *List, Sequence* consists of operations involving the `list` type and Why3's `sequence`, `array31`, `array32`, and `array63`.

- *Set, Map, Bag* consists of operations whose types involve finite map, multiset, finite set, predicate-based set, and hash-table.

- *Tree, String, Matrix* consists of operations whose types involve Why3's built-in binary tree, string, and matrix.

- *Memory* consists of operations whose types involve Frama-C's memory encoding.

- *Custom Datatype* consists of operations whose types involve any datatype not provided by the system library but defined by the verification projects.

The metric *depth* is the height of the abstract syntax tree of the VCs, in the standard $\lambda$-calculus representation with all arguments of every function application represented as siblings.

## I    INTERSECTION ANALYSIS OF NTP AND HAMMER CAPABILITIES

To understand whether neural and symbolic approaches overlap or diverge in their capabilities, we analyze the intersection between the union of all problems solved by NTP models and the union of all problems solved by hammers. Table 7 presents the results. The results reveal a complementarity: many verification conditions are solved exclusively by one method or the other. This confirms that NTPs and hammers leverage distinct reasoning mechanisms and that neither approach is a subset of the other, highlighting the potential for hybrid solutions.

Table 7: The number of problems solved by both hammers and NTP models, only by hammers, and only by NTP models.

| Category | Common | Hammer only | NTP only |
|---|---|---|---|
| Algorithm | 1 | 3 | 1 |
| Data Structure | 3 | 7 | 0 |
| Calculation | 3 | 5 | 2 |
| Engineering | 5 | 2 | 0 |
| Competition | 0 | 3 | 3 |
| Function | 6 | 19 | 0 |
| Memory | 1 | 17 | 1 |
| Loop | 3 | 16 | 0 |
| Invalid Arg. | 5 | 15 | 0 |
| Total | 27 | 87 | 7 |

## J    IDENTIFYING VCS IN COQSTOP AND FVEL

In order to support the numbers given in Tab. 1, this section describes our approach to identifying VCs in the CoqStoq benchmark (Thompson et al., 2025) and FVEL (Lin et al., 2024). CoqStop's test set contains 10,396 theorems from 12 Rocq projects; FVEL's test set contains 1967 cases.

**CoqStop**    CoqStop's VCs are predominantly drawn from CompCert, which accounts for over 58% of the test set, while other verification-related projects constitute no more than 6%. Therefore, we focus solely on CompCert. In CompCert, the tactics and other constructs that are relevant to program analysis and VC generation are `TransfInstr`, `UseTransfer`, `monadInv`, `step_simulation`, `exploit`, and `match_states`. Among the CompCert VCs in CoqStop, only 1,325 cases involve these tactics, accounting for 12.7% of the total test set. Including other projects that may involve VCs (at most 6%), the total proportion would not exceed 20%.

**FVEL**    All of FVEL's test cases are extracted from seL4. seL4's VCs are generated using the tactics vcg, wp, and wpsimp. Based on the test case list provided by FVEL, we analyzed cases whose proofs contain these tactics and found only 328. Therefore, the proportion of VCs in FVEL does not exceed $328/1967 < 17\%$.

## K    THE USE OF LARGE LANGUAGE MODELS (LLMS)

We have used LLM as a writing aid to assist with fluency and grammatical checking.

