# OpenReview forum: "Neural Theorem Proving for Verification Conditions: A Real-World Benchmark"
_ICLR.cc/2026/Conference — ICLR 2026 Poster_

### Official Review · Reviewer_MX3g · 2025-10-25

**Soundness:** 4
**Presentation:** 3
**Contribution:** 3
**Rating:** 6
**Confidence:** 2

**Summary:**

The paper introduces NTP4VC, a new task and benchmark that targets neural theorem proving of verification conditions (VCs)—arguably a core bottleneck in automated program verification. The authors (i) extract VCs from real-world projects via industrial pipelines (Why3/Frama-C), (ii) translate them with ~2.4K expert-written mapping/rewriting rules into three ITPs (Lean, Isabelle, Rocq), (iii) complicate the VCs by erasing helper annotations to make proofs harder while remaining provable, and (iv) evaluate LLM provers and hammer baselines. The released benchmark contains 672 VCs spanning pearls (algorithms/data structures) and real C verification (Contiki OS, Linux scheduler, etc.), with difficulty balanced so Why3’s strongest ATP tactic hovers around ~20% success. Across languages, modern LLM provers achieve <4% pass@8 while Isabelle/Coq hammers perform better but still modestly (up to ~15% pass@1).

**Strengths:**

1. Problem focus is well-motivated. VC theorem proving sits at the heart of verification workflows and is less explored by NTP than math benchmarks. By targeting VCs directly (rather than only annotation synthesis), the paper addresses a real bottleneck and usefully complements recent annotation-centric efforts such as Laurel and DafnyBench.

2. Extracting from Why3/Frama-C, translating to Isabelle/Lean/Rocq, and sourcing from Contiki/Linux bring the setting closer to practice than e.g., purely math (miniF2F/PutnamBench) or toy code-property suites.  The two halves (pearls vs. real C) and per-category balancing around a target ATP pass rate are thoughtfully constructed; the complication step measurably increases hardness.

3. Results convincingly show VCs are qualitatively harder for current NTP models than math/formal benchmarks.

**Weaknesses:**

1. Concerns regarding translation correctness: The benchmark depends on ~2.4K hand-written translation/rewriting rules and occasionally axiomatizes termination when Isabelle/Rocq can’t discharge it. There is no formal correctness proof of the translation or quantitative post-hoc validation beyond syntax checks/cross-review, leaving risk of subtle semantic drift or unsoundness through added axioms. A minimal mechanized proof of semantics-preservation for a core calculus would be beneficial.

2. Contamination analysis is qualitative. Given that “engineering” cases show unusually high NTP success relative to others, please run decontamination (code/proof snippet search against common training corpora) to support the low-contamination claim.

**Questions:**

1. From >5.3k extracted VCs, 672 are “carefully selected,” but criteria are only qualitatively described. Can you please release the full pool with metadata (size, quantifier alternation, AST depth, symbol counts, dependency graph) and the sampling procedure?

2. For a representative subset, can you show end-to-end proofs exist in at least one target ITP (perhaps with generous hammer help), confirming that difficulty stems from complexity rather than unprovability introduced by erasing?

---

> ### Author Response · Authors · 2025-11-22
>
> We genuinely appreciate the reviewer's positive assessment and valuable feedback on our work, and are grateful for the reviewer's acknowledgment of NTP4VC as addressing a well-motivated problem that sits at the heart of verification workflows.
> ### Weakness 1. Concerns regarding translation correctness
> We agree that a mechanized proof of semantics-preservation would provide the strongest guarantee for our translation pipeline. However, this would require mechanized semantic formalizations of Why3, Isabelle, Lean, and Rocq all within one proof assistant (or at least pairs of Why3-Isabelle, Why3-Lean, and Why3-Rocq within multiple proof assistants). Furthermore, formalizing the core calculi alone is insufficient — our translation relies on extended theories for integer, list, map, set, and other data structures, which also need to be formalized. This is an extremely challenging undertaking for which no existing work provides the necessary infrastructure, and developing it from scratch would be prohibitively expensive — the formalization of Rocq's core calculus alone ([MetaRocq](https://arxiv.org/pdf/2502.15500)) comprises over 200K lines of Rocq code. Finally, proving semantic preservation itself is also exceptionally challenging. For reference, [CompCert](https://inria.hal.science/hal-01238879/) — a certified compiler with semantic preservation proofs — required ~6 person-years of effort and ~100K lines of Rocq code. As another example, [Cake ML](https://cakeml.org/jfp19.pdf) also costs ~100K lines of HOL4 code.
>
> Such an effort would be prohibitively costly; instead, a more practical approach is to incrementally complete mechanized proofs for the 672 benchmark cases as our future work.
>
> ### Weakness 2. Missing decontamination (code/proof snippet search)
>
> We thank the reviewer for raising this important point. To assess potential training contamination, we performed a decontamination analysis by searching for overlapping **13-gram** sequences between our benchmark and [**Proof-Pile 2**](https://huggingface.co/datasets/EleutherAI/proof-pile-2), a 55-billion-token corpus of mathematical and theorem-proving documents. The **13-gram threshold** is an empirically established standard adopted in prior work such as *SlimPajama* for detecting content overlap.
>
> No overlapping 13-grams were found, supporting our claim of minimal training contamination.
>
> | Dataset      | Document Count | n-gram Size | Overlapping n-grams |
> |---|---|---|---|
> |Proof-Pile 2|11,314,233 | 13 | 0 |
> |Ours           |2,016          | 13 | 0 |
>
> ### Q1. Release the metadata and the sampling procedure.
>
> We thank the reviewer for this important point. A dedicated subsection titled 'VC Selection Process' has been added in the revised draft to detail the sampling procedure. We release the meta data of [all extracted VCs](https://drive.google.com/file/d/1sYQlRfIz8gXBdfl0jpRZCxJGRv54igZ8/view) and [benchmark cases](https://drive.google.com/file/d/1Y8EdyaO_i0wARVC_cVJND4lDlhXIKed2/view).
>
> To summarize the sampling procedure: We first ran Why3's automated prover AL3 to assess the difficulty of each VC case. Hard VCs that cannot be solved by AL3 within 10 minutes are considered more valuable and thus prioritized for inclusion in our benchmark. We initially selected ~900 hard VCs. Since we use AL3 as our difficulty baseline and aim to maintain its pass rate at 20-30%, we correspondingly select 300 easy VCs that AL3 can solve. From these ~1.2K candidates, we manually inspected each VC's expression, source program, specifications, and annotations, selecting those that represent meaningful and commonly encountered verification tasks. This process resulted in the final 672 benchmark cases.
>
> ### Q2. For a subset, show end-to-end proofs exist.
>
> We randomly select proof goals from the hard VCs that remain unproven by both models and hammers. Due to the limited rebuttal period, we have completed manual proofs for 16 cases in Isabelle, which are released at [here (see proven_goals.lst)](https://drive.google.com/file/d/1kQQ4m2xQK3o4DCRvi2WIRRLePpUo4jCY/view). We are actively working on additional proofs and will continue to update this link.

---

### Official Review · Reviewer_B7qC · 2025-10-31

**Soundness:** 3
**Presentation:** 4
**Contribution:** 3
**Rating:** 4
**Confidence:** 4

**Summary:**

The authors propose a benchmark for proving verification conditions. They extract these benchmarks from a combination of real world code, where the VCs are extracted using existing industrial tools, and are then simplified massively by removing hints like asserts and lemmas. The authors then evaluate existing LLMs, NTPs and Hammers.

**Strengths:**

1. The work is novel to the best of my knowledge, and addresses a fundamental gap: automating program verification for real-world complex programs.
2. Thorough evaluation of models and NTPs on the generated datasets. I also appreciate the datasets being created for different languages.

**Weaknesses:**

1. Lack of details on how the 5.3K VCs get filtered down to 627: I'm not sure why this was done, or what were the criteria for eliminating majority of the VCs. Were they too similar to each other, making them redundant? Or was it some deeper issue? Regardless, the elimination process was not described in the paper. Furthermore, out of the 5.3K, how many were extracted from the real-world programs and how many were extracted from the pearls of programs?
2. The pearls of programs: Does the inclusion of these pearls of programs not contradict the real-world claim of the title of the paper? I understand the need for diverse VCs, but the pearls seem more as an exercise in niche settings rather than VCs that occur frequently in realistic programs.
3. Discussion centered around improving NTPs: I would like a discussion that focuses on how NTPs could improve on such VCs. I assume that these VCs would be difficult to prove even for humans. Even if one would try to perform RL, the initial accuracies of NTPs are not enough to provide straightforward gains. How could one try to improve NTPs on such complex benchmarks?

**Questions:**

See weaknesses. Also:
1. Have you thought about composing different VCs to try and create even more complex benchmarks?
2. How easy is it to add more projects to the set of real-world verified programs?
3. Can you elaborate on your statement in lines 403 -- 408? I am not sure I really understand why one category performs better than the other.

---

> ### Author Response · Authors · 2025-11-22
>
> We thank the reviewer for the thoughtful and constructive feedback, and for recognizing the novelty and significance of our work in advancing automated verification for real-world programs. We address each concern below.
> ### Weakness 1. How and why the 5.3K VCs get filtered down.
>
> We acknowledge that the original paper did not sufficiently explain this process — this was an oversight on our part, and we appreciate the reviewer for pointing it out.
> We have now added a detailed description in the subsection “VC Selection Process.” Summarily, the filtering was driven by both practical and scientific considerations:
>
> 1. Evaluation Efficiency. A smaller benchmark enables more efficient model evaluation. Evaluation metrics such as pass@n require running models multiple times (often up to hundreds of iterations). Under this setting, benchmark size significantly impacts evaluation cost. Recent studies have advocated for small but high-quality benchmarks, [Vivek et al](https://arxiv.org/abs/2309.08638), [Perlitz et al](https://arxiv.org/abs/2308.11696), [Li et al](https://aclanthology.org/2025.emnlp-main.716).
> 2. Difficulty. Many extracted VCs can be solved by existing automated provers (ATPs), making them less meaningful benchmark targets. We therefore filter out most of these cases. This leaves ~1.6k hard VCs.
> 3. Relevance. Some VCs are generated from lemmas instead of programs. They are removed, leaving ~1k hard VCs.
> 4. Further extensibility. We intentionally reserve some hard VCs for potential future training use.
>
> Out of the 5.3K, ~2.3K are extracted from the real-world C projects, and ~3.0K are from the pearls of programs. Although our benchmark selects only 672 VCs, all our extracted VCs will be released.
> ### Weakness 2. Pearls contradict the real-world claim; they are niche.
> We respectfully disagree. Programs in Pearls are minimal implementations of well-known algorithms and data structures, not niche exercises. They capture **core verification challenges** in real-world systems, making their inclusion essential for a verification benchmark. They appear sparsely in real-world projects, making it difficult to achieve comprehensive coverage in a small benchmark if there is not a specific concentration like Pearls of Programs.
>
> Furthermore, the VCs are extracted by Why3, an industry-grade verification framework, ensuring alignment with real-world verification forms.
> We organize pearls and real C programs into separate categories, each containing sufficient cases (compared to popular benchmarks like miniF2F and PutnamBench), allowing evaluation of algorithmic and engineering domains.
> Finally, while we acknowledge that algorithm verification can be competition-driven or application-driven, the inclusion of Pearls, given all the above, does not undermine the real-world relevance of our benchmark.
>
> ### Weakness 3. How could NTP improve on complex VCs?
> We recognize this as a valid concern, but we view it as a challenge for the field, not a weakness of our benchmark. AI capabilities in reasoning and formal methods are advancing rapidly. While we cannot guarantee that models will succeed in VC proving, our benchmark identifies an important direction for AI-driven program verification and provides the groundwork for measuring progress in this area.
>
> When mathematical proof benchmarks such as miniF2F were introduced four years ago, there were similar doubts about whether large language models were suitable for such tasks. Yet within a few years, these benchmarks have approached saturation. **Similar rapid progress has been observed across many recent reasoning benchmarks**—for example, performance on HLE (https://agi.safe.ai) has risen from 9.4% at its release in January 2025 to 37.5% today, and on SWEBench (https://www.swebench.com) from 6% at release in 2024 to 74% currently. In each case, initial skepticism quickly gave way to significant advances once the community focused attention on the challenge. We expect our benchmark to play a similar **catalytic role** for program verification.

---

> ### Author Response · Authors · 2025-11-22
>
> ### Q1. Composing VCs to create more benchmarks.
> This is a good idea! We think composing VCs can be a valuable approach for generating training data. Regarding our benchmark construction, we prioritize VCs with authentic semantic grounding in real programs and algorithms. Artificially composed VCs may lack the semantic structure found in genuine program verification scenarios, limiting their value for evaluation. We do have a plan to explore composition for training data in future work.
>
> ### Q2. How easy to add more projects to the set of real-world verified programs?
> Our pipeline has mapped all the Frama-C dependencies into Isabelle/Lean/Rocq. This means any C project verified by Frama-C’s WP engine (its standard backend for program verification) can be processed by our pipeline to export VCs automatically into Isabelle/Lean/Rocq. The labor cost is to set up our pipeline and run commands.
>
> ### Q3. Elaborate on lines 403 -- 408
> The lines 403-408 were poorly written. We have rewritten the paragraph in the updated draft. Summarily, language models and hammers have different working mechanisms. Models tend to perform better in domains where they have been exposed to relevant knowledge during training. Since cases in the Engineering category stem from common programming patterns and implementation optimizations that models have extensively encountered during pre-training, this may explain why models perform better in this category.
>
> We sincerely thank the reviewer for their valuable feedback. We have clarified the VC selection process, justified the inclusion of Pearls of Programs, and addressed all raised questions. These revisions have strengthened both the clarity and completeness of our work.
>
> Regarding the concern about the current models’ difficulty with the benchmark, we view this as an opportunity rather than a weakness.
> We have manually proved several VCs—specifically those noted by reviewers c1Bg and MX3g—to demonstrate their provability and to show that the challenge lies in model capability rather than in benchmark design.
>
> We believe our benchmark fills a crucial gap in enabling systematic study of AI-driven program verification and will serve as a solid foundation for future research in this emerging direction.

---

> > ### Comment · Reviewer_B7qC · 2025-11-27
> >
> > I thank the authors for their responses and for clarifying my issues! Just to be clear, I do not view the difficulty of the benchmark as a weakness! But I do feel that a paper proposing a benchmark should also include a discussion on the current weaknesses and some insights on how to overcome them. Part of it is in the error analysis discussion, but a discussion centered around future possibilities for advancement would be really helpful from the perspective of helping others improve NTPs on your benchmark.
> >
> > I'll increase my score to a 6.
> >
> > PS. I think Tables 5 and 6 would be improved by splitting accuracies over real C programs v/s the pearls of programs. Specifically, in Table 5, each cell could be P@K (P@K over C programs / P@K over Pearls). For Table 6, you could insert aggregate rows for the subcategories for Real C programs and for the Pearls of Programs.

---

### Official Review · Reviewer_c1Bg · 2025-10-31

**Soundness:** 2
**Presentation:** 3
**Contribution:** 2
**Rating:** 4
**Confidence:** 3

**Summary:**

The authors introduce NTP4VC, a novel benchmark targeting neural theorem proving (NTP) for program verification conditions (VCs). Unlike prior datasets focused on mathematical or auxiliary lemmas, NTP4VC consists of real-world VCs automatically extracted from industrial projects using mature verification pipelines. The benchmark covers multiple languages and a diverse range of verification scenarios, aiming to reflect practical challenges in automated program verification. Experimental results show that existing neural and automated provers perform poorly on NTP4VC.

**Strengths:**

1. **Benchmark contribution:**

The authors introduce a new dataset specifically addressing VC proving, which is a practical bottleneck in automated program verification.

2. **Industrial relevance:**

VCs are extracted from real-world industrial projects using established verification pipelines (Why3, Frama-C), increasing practical relevance.

3. **Multi-language and scenario coverage:**

The dataset spans multiple proof assistant languages and aims to cover a variety of verification scenarios.

**Weaknesses:**

1. **Limited dataset novelty:**

Although the focus on VCs is clear, the novelty is moderate. Previous datasets (e.g., CoqStop, FVEL, CoqGym) have also included verification-related theorems, even if the VC proportion was lower. The paper should better highlight what fundamentally differentiates NTP4VC—beyond just the percentage of VCs.

2. **Lack of technical contribution:**

The work mainly involves dataset construction and benchmarking; there is no new technical advance.

3. **Provability not manually verified:**

While VCs are generated using automated tools, there is no evidence of manual or sampled verification to guarantee their provability. This raises the risk that some VCs may be invalid or unprovable due to the toolchain.

4. **No quantitative analysis of VC diversity:**

The paper focuses on diversity of program sources, but does not quantitatively analyze the types of generated VCs. It is unclear whether the dataset covers a broad range of VC properties or is dominated by a few repetitive types. A breakdown of VC categories/properties would strengthen the empirical evaluation.

**Questions:**

See Weaknesses

---

> ### Author Response · Authors · 2025-11-22
>
> Thank you for your constructive feedback. We appreciate that you recognized our motivation, industrial relevance, and multi-language coverage. Below, we provide additional clarifications and context regarding the four weaknesses you identified, which we hope will help convey the novelty and significance of our contribution more clearly.
> ### 1. Limited dataset novelty
> Thank you for the comment. We agree that previous datasets (e.g., CoqGym, FVEL, CoqStop) contain verification-related theorems. However, NTP4VC is the first benchmark designed specifically and exclusively for verification conditions (VCs), with **systematic balancing of difficulty levels and diversity across categories**. This focus represents a clear shift from prior datasets where VCs appeared only incidentally.
>
> As you noted in our strengths, **multi-language coverage** represents a distinctive feature of our benchmark. Indeed, our benchmark is **the first to support Lean, Rocq, and Isabelle**, enabling **fair and consistent cross-system evaluation** of NTPs.
>
>
> We believe these aspects together establish NTP4VC as a substantially new and complementary contribution to the field.
> ### 2. Lack of technical contribution
> Thank you for this valuable comment. We apologize that we may not have adequately emphasized our technical contributions. In fact, our work involves substantial technical development behind the dataset construction. Specifically, we created a comprehensive methodology for extracting verification conditions (VCs) from real verification projects and a rule-based translation framework that converts these VCs into multiple interactive theorem provers (ITPs).
>
> We have released a complete and reproducible pipeline implementation, consisting of ~800 translation rules per ITP (≈2400 total), ≈4K lines of Python translation code, supporting libraries for Isabelle (≈500 lines), Lean (≈160 lines), and Rocq (≈200 lines), and a ≈200-line Why3 patch. All components were carefully written by experts in theorem proving and program verification, ensuring correctness and maintainability. These efforts together represent a significant and nontrivial engineering contribution to the AI4Math and AI4Verification communities
>
> We have added these details to Appendix E for better clarification.
>
> Finally, we would respectfully draw attention to technical contributions that other reviewers have recognized:
> - Reviewer eJD5’s strength 2: “The benchmark construction is based on high-quality expert-written rules, which is reliable and one-time effort”
> - Reviewer MX3g’s strength 2: “Extracting from Why3/Frama-C, translating to Isabelle/Lean/Rocq, …”
>
> ### 3. Provability not manually verified
>
> We acknowledge that our benchmark does not include full manual verification for every case. However, the provability is strongly supported by the rigor of our data sources. All our benchmark cases are extracted from peer-reviewed formal verification projects. The sources of the benchmark cases are all formally verified. Admittedly, this cannot rule out the possibility of unprovable cases due to bugs in Frama-C, Why3, and our translator. Nonetheless, the application of Frama-C and Why3 at least provides substantially stronger trustworthiness guarantees than other popular benchmarks in neural theorem proving (e.g., MiniF2F, PutnamBench), which rely solely on human review and have been found to contain many errors as reported by [Ospanov](https://arxiv.org/abs/2511.03108) and [Cao](https://arxiv.org/abs/2506.11487).
>
> Besides these strong reasons supporting provability, we have conducted two additional validations to further ensure the soundness of our benchmark. To address your concerns.
> We have run the counterexample generator [Nitpick](https://link.springer.com/chapter/10.1007/978-3-642-14052-5_11) during this rebuttal session, showing no simple counterexample found on our benchmark cases, indicating none of them are trivially unprovable.
>
> We randomly select proof goals from the hard VCs that remain unproven by both models and hammers. Due to the limited rebuttal period, we have completed manual proofs for 16 cases in Isabelle, which are released at [here (see proven_goals.lst)](https://drive.google.com/file/d/1kQQ4m2xQK3o4DCRvi2WIRRLePpUo4jCY/view). We are actively working on additional proofs and will continue to update this link during the remaining rebuttal period.

---

> ### Author Response · Authors · 2025-11-22
>
> ### 4. No quantitative analysis of VC diversity. The dataset is dominated by a few repetitive types.
>
> Thank you for this helpful reminder, we have conducted a quantitative analysis of VC diversity and presented it in the updated paper draft (Table 4). A simplified table is presented as follows,
>
> |Category | # of cases | average # of operations | 25th, 75th percentile|
> |---|---|---|---|
> |Integer Artih| 645 | 60.1 | 13, 68|
> |Non-linear Arith| 106 | 11.4 | 2, 13|
> |List, Sequence| 234 | 47.2 | 8, 60.5|
> |Seq, Map, Bag| 67 | 46.6 | 9, 48|
> |Tree, String, Matrix| 33 | 53.5 | 12, 84|
> |Memory| 336 | 36.2 | 13, 36|
> |Custom datatype| 242 | 96.0| 15, 102|
>
> The results demonstrate that VCs span diverse operations and data structures, including: integer arithmetics, non-linear arithmetics, list, sequence, set, map, multiset, tree, string, matrix, memory, and custom datatypes, with a broad distribution of VC sizes within each category. This diversity of the notions involved in the benchmark cases demonstrates that the cases are highly unlikely to be dominated by a few repetitive types.
>
> Overall, thank you for your thoughtful comments and for highlighting areas that could be clarified further. Your feedback has helped us improve the clarity and completeness of the paper. We hope our rebuttal has addressed your concerns and believe our responses demonstrate the novelty, rigor, and practical value of our work.

---

### Official Review · Reviewer_eJD5 · 2025-11-09

**Soundness:** 4
**Presentation:** 4
**Contribution:** 3
**Rating:** 8
**Confidence:** 5

**Summary:**

This paper propose a challenging and real-world program verification benchmark, theorem proving for verification conditions (VCs). Different from most theorem proving benchmarks, VCs are the primary (and transient) intermediate representations of program verification in the real world. This work first collects VCs by instrumenting a popular program verifier, Why3, and then converting them into three popular interactive theorem provers (ITPs), i.e., Isabelle, Lean, and Rocq,  based on manually designed rules (around 800 for each ITP). The verification tasks span two categories: 1) small challenging puzzles that involve non-trivial data structures and algorithms, and 2) C programs from real-world projects. The experimental evaluations show that neural theorem provers (NTP) achieve less than 4% pass rate, which is consistently worse than classic ATPs like CoqHammer/Sledgehammer. The new benchmark remains challenging for both ITPs and ATPs, motivating technical innovations in both.

**Strengths:**

- Constructing a real-world program verification benchmark using VCs is well-motivated and the paper is well-written.
- The benchmark construction is based on high-quality expert-written rules, which is reliable and one-time effort.
- The NTP4VC benchmark strikes a good balance of two categories of challenging program verifications, specifically, small programming puzzles and real-world projects. Furthermore, the transient nature of VCs make potential contamination less of a concern.
- Experimental evaluations show the clear gap as well as promises of neural theorem provers, motivating future follow-up works.

**Weaknesses:**

- compared to programs and annotations at the source code level, VCs and relevant proofs are perhaps too low-level and less suitable for LLMs to generate proofs, since human experts rarely write proofs for VCs. Although NTP4VCs is an interesting and challenging benchmark, but may not be necessary to tackle directly by LLMs. The cost is also prohibitive, as indicated by the limited evaluation with GPT-o4-mini-high.
- three themes of errors (presented in sec 5.2) are well expected; having a quantitative analysis will be more helpful.

**Questions:**

How large are the translated theorems in ITPs? Is there always a single proof goal? Are they human readable?

The results presented in Table 5 are really interesting. Besides comparing the absolute number of solved theorems, are they complementary or roughly a subset of another?

---

> ### Author Response · Authors · 2025-11-22
>
> We sincerely thank the reviewer for the thoughtful and detailed feedback. We are pleased that the reviewer found the soundness and presentation excellent, and appreciated the well-motivated construction and contributions of our NTP4VC benchmark.  Below, we present our detailed answers to the raised questions and comments.
> ### Weakness 1. VCs and relevant proofs are low-level and less suitable for LLMs to generate proofs, since human experts rarely write proofs for VCs.
> We agree with the reviewer that VCs are based on direct encodings of program operations and thus appear low-level. To provide additional context, experts in programming languages and verification do occasionally write manual proofs for VCs, as seen in verified systems such as [seL4](https://doi.org/10.1145/1629575.1629596), [CertiKOS](https://www.usenix.org/conference/osdi16/technical-sessions/presentation/gu), and [Compcert](https://inria.hal.science/hal-01238879).
> In these projects, VCs are difficult and cannot be solved by automated provers (ATPs) alone;
> human experts manually analyze proof goals, transform and decompose them into simpler subgoals tractable for ATPs. This process is precisely what NTP4VC aims to capture — enabling LLMs to learn such transformation and decomposition, thereby extending ATPs to previously intractable VCs.
>
> ### Weakness 2. Having a quantitative analysis (for sec 5.2) will be more helpful
> This is an excellent suggestion. We have added related statistics and examples to our updated manuscript (Section 5.2 and Appendix G). Summarily, semantic and pragmatic confusion is the most common type of error. For example, more than 64% Lean proofs generated by Goedel-Prover-V2-32B contain repetitive and meaningless tactic applications on our benchmark. Syntax errors are also common in the generated proofs, with at least 24% Isabelle proofs containing syntactic errors. We further collect statistics about hallucination of non-existent entities by checking error messages, where at least 9% proof failures are directly caused by these undefined facts or constants.
> ### Q1. How large are the translated theorems in ITPs?
> The scale of the translated proof goals varies from dozens of atomic terms to more than a thousand, averaging 653 terms, with 25th, 50th, and 75th percentiles of 158, 350, and 754. Details can be found in [benchmark_meta_data.csv](https://drive.google.com/file/d/1Y8EdyaO_i0wARVC_cVJND4lDlhXIKed2/view?usp=sharing).
>
> > Is there always a single proof goal?
>
> Yes, each benchmark case consists of exactly one verification condition (VC), and in the context of our work, one VC corresponds to one proof goal.
>
> > Are they human readable?
>
> The readability of the proof goals varies by scale. For small proof goals, the statements are human-readable, and one can typically infer from the proof goal the intended properties that the source program should satisfy. With some effort, humans can also identify the lemmas, induction principles, and other proof techniques needed to prove the goal. However, for the larger proof goals that predominate in our benchmark, direct comprehension is challenging. A difficulty is that humans struggle to process the sheer volume of expressions and to develop a holistic understanding. Nonetheless, after applying (interactive) tactics and proof automation tools to transform and decompose these goals, humans can still partially understand them, though associating them back to the original source program becomes difficult after such transformations.
>
> ### Q2. Are hammers and NTPs complementary or roughly a subset of another?
> We analyzed the intersection between the union of problems solved by LLMs and the union of problems solved by hammers. The results, shown in the table below, confirm that the two approaches are complementary.
>
> |Category | Common | Hammer uniquely solved | LLM uniquely solved |
> |---|---|---|---|
> | Algorithm| 2 | 4 | 4 |
> | Data Structure | 5 | 6 | 3 |
> | Calculation| 6 | 7 | 2 |
> | Engineering| 6 | 1 | 14 |
> | Competition | 0 | 3 | 1 |
> | Function | 6 | 12 | 3 |
> | Memory| 7 | 8 | 4 |
> | Loop | 4 | 14 | 2 |
> | Invalid Arg.| 6 | 10 | 1 |
>
> Furthermore, within each category, both models and hammers solve some problems that the other cannot, encouragingly demonstrating that models can outperform hammers at least on certain problems. On the other hand, it also reveals that models have not yet learned to effectively leverage hammers for VC proving --- ideally, models should utilize hammers when appropriate, making the hammer-solvable set roughly a subset of the model-solvable set.

---

### Author Response · Authors · 2025-12-04
**Authors' Response Summary**

We sincerely thank all reviewers for their time and constructive feedback. We are encouraged by the reviewers' recognition of NTP4VC as a "well-motivated" (eJD5, MX3g) and "novel" (B7qC) benchmark that addresses a "fundamental gap" (B7qC) in applying AI to real-world program verification with strong “industrial relevance” (c1Bg).

**We are particularly grateful for the active engagement during the rebuttal period, and we deeply appreciate Reviewer B7qC’s decision to raise their score to 6 following our clarifications.**

We have comprehensively revised the manuscript in accordance with the reviewers' valuable suggestions. The revisions and responses to the reviewers’ questions are summarized as follows:

1. **More Validation for Provability (Responding to Reviewers c1Bg, MX3g).**\
To address concerns regarding the provability of the generated VCs and the correctness of our translation:
- We have selected 16 VCs unsolved by both models and hammers as a representative subset and manually provided their mechanically verified proofs. The proofs are released [here](https://drive.google.com/file/d/1kQQ4m2xQK3o4DCRvi2WIRRLePpUo4jCY).
- We conducted a counterexample check using Nitpick, which found no trivial counterexamples.

2. **Expanded Quantitative Analyses (Responding to Reviewers eJD5, c1Bg).**
- *Intersection Analysis:* Our additional data analysis demonstrated that NTPs and Hammers are complementary —  NTPs uniquely solve 14 problems in the Engineering category where Hammers fail, highlighting the unique value of learning-based approaches.\
- *Error Analysis:* We provided lower-bound statistics on failure modes, showing that at least 24% of Isabelle proofs fail due to syntax errors and 9% due to hallucinations of undefined entities.
- *Diversity Analysis:* We added a breakdown analysis of VC operations (arithmetic, maps, memory, etc.) to confirm the diversity of the benchmark.

3. **Detailed Methodology and More Metadata (Responding to Reviewers B7qC, MX3g).**
- *VC Selection:* We added a dedicated subsection in Section 5 to detail our filtering process (from 5.3K to 672 cases) based on evaluation efficiency, difficulty (filtering out easy ATP-solvable cases), and relevance.
- *Decontamination:* We performed a 13-gram overlap search against Proof-Pile 2, confirming no contamination was found.
Metadata: We have released full VC metadata as required by the reviewer MX3g.

We believe NTP4VC's most significant contribution is being the first to direct the emerging field of neural theorem proving for program verification toward a key industrial bottleneck: VC proving, while providing the first benchmark drawn from real-world verification projects like Linux and Contiki-OS. It is also multi-lingual across Lean, Rocq, and Isabelle. We hope the additional evidence and revisions provided during the rebuttal have addressed the reviewers' concerns.

Thank you again for your consideration.

---

### Meta-Review · Area_Chair_qy3x · 2026-01-06

**Summary:**

The paper proposes NTP4VC, a novel multi-lingual benchmark (Isabelle, Lean, Rocq) for Neural Theorem Proving focused on Verification Conditions (VCs) derived from real-world projects like Linux and Contiki-OS. The reviewers unanimously appreciated the strong motivation, the industrial relevance of using Why3/Frama-C pipelines, and the significant engineering effort involved in the rule-based translation. Concerns primarily focused on the provability of generated VCs, the novelty compared to existing datasets, the specifics of the filtering process, and potential contamination.

**Reviewer Concerns:**

The rebuttal successfully addressed the majority of the reviewers' concerns.
1. **Provability & Correctness (c1Bg, MX3g):** The authors addressed concerns about the validity of the VCs by providing manually verified proofs for a representative subset of "hard" problems and running Nitpick to check for counterexamples.
2. **Dataset Statistics & Diversity (eJD5, c1Bg):** The authors provided the requested quantitative analysis of VC operations and an intersection analysis between NTPs and Hammers, demonstrating the diversity and complementary nature of the benchmark.
3. **Filtering Process (B7qC, MX3g):** The authors clarified the selection criteria (efficiency, difficulty, relevance) for narrowing down the dataset from 5.3k to 672 cases and released the full metadata.
4. **Contamination (MX3g):** The authors performed a 13-gram overlap search against Proof-Pile 2, confirming no contamination.

The only outstanding point is the lack of a fully mechanized proof for the translation rules themselves (raised by **MX3g**), which the authors argued is prohibitively expensive for the current scope; this explanation is reasonable given the scale of the contribution.

**Reviewer Scores:**

- **eJD5: 8**
The reviewer was already positive (score 8) regarding the benchmark's motivation and quality, and the rebuttal satisfactorily answered their specific questions about proof size and readability.

- **c1Bg: 6**
The reviewer initially rated 4 due to concerns about unverified provability and lack of diversity analysis; since the authors provided manual proofs for hard cases and detailed diversity statistics, these concerns are resolved.

- **B7qC: 6**
The reviewer explicitly raised their score to 6 during the discussion phase after the authors clarified the filtering process and justified the inclusion of "Pearls" of programs.

- **MX3g: 6**
While the authors performed the requested decontamination check and provided manual proofs for a subset, the lack of a formal correctness proof for the translation rules themselves—though practically justified—remains a limitation that prevents a higher score.

---

### Decision · Program_Chairs · 2026-01-26

Accept (Poster)